# The increasing burden of group B *Streptococcus* from 2013 to 2023: a retrospective cohort study in Beijing, China

Yingxing Li,[1,2,3] Wenhang Yang,[1,2] Yi Li,[1,2] Kexin Hua,[1,2] Ying Zhao,[1,2] Taie Wang,[1,2] Lingli Liu,[1,2] Yali Liu,[1,2] Yao Wang,[1,2] Wenjing Liu,[1,2] Li Zhang,[1,2] Renyuan Zhu,[1,2] Shuying Yu,[1,2] Hongli Sun,[1,2] Hongtao Dou,[1,2] Qiwen Yang,[1,2] Yingchun Xu,[1,2] Lina Guo[1,2]

**ABSTRACT**  Group B *Streptococcus* (GBS) is a leading pathogen responsible for fatal infections in newborns primarily due to vertical transmission from colonized mothers. Cases of invasive GBS infections in adults have also increased and attracted attention recently. To comprehensively understand the evolving burden of vaginal GBS carriage in pregnant and non-pregnant women, as well as the trends in invasive GBS diseases and antibiotic resistance in China, we conducted a retrospective study using data from a large tertiary hospital in Beijing from 2013 to 2023. Over the past decade, improvements in GBS screening methods for pregnant women have significantly increased the GBS recovery rate. The detection rate of GBS and its proportion among vaginal pathogens have shown a gradual increase in GBS colonization in both pregnant and non-pregnant women. An analysis of vaginal pathogen composition revealed variations in GBS prevalence across different age groups, as well as a potential competitive relationship between GBS and *Enterococcus faecalis* in the vaginal environment. Additionally, we analyzed 165 invasive GBS cases, including three in newborns. The incidence of invasive GBS cases has risen since 2016, particularly among individuals over the age of 40. The 5,858 GBS isolates exhibited notably high resistance rates to erythromycin (72.2%), clindamycin (60%), and levofloxacin (50.1%), with 30.8% classified as multidrug-resistant. Importantly, invasive GBS strains exhibited a higher resistance rate to levofloxacin (61.2%) compared to colonizing strains (49.8%). This study highlights the importance of continuous screening and monitoring for GBS, especially given the concerning antibiotic resistance rates of GBS.

**IMPORTANCE**  Group B *Streptococcus* (GBS) is an important pathogen that commonly causes infections in newborns and the elderly. This retrospective study provides a comprehensive analysis of GBS strains isolated from a large tertiary hospital in Beijing between 2013 and 2023, revealing an increasing colonization rate of GBS in both pregnant and non-pregnant women. Analysis of vaginal pathogens indicates a growing proportion of GBS among vaginal pathogens. Additionally, the high resistance rates of GBS to erythromycin, clindamycin, and levofloxacin, as well as the prevalence of multidrug resistance, are issues that merit attention. We also examined the differences in resistance rates of GBS strains from various sample types, finding that the levofloxacin resistance rate in GBS strains causing invasive infections was significantly higher than in colonizing strains. This study provides new data and insights for clinical research on GBS.

**KEYWORDS**  group B *Streptococcus*, vaginal GBS carriage, invasive GBS infection, antibiotic resistance

Group B *Streptococcus* (GBS), also known as *Streptococcus agalactiae*, is a faculta-tive anaerobic Gram-positive coccus. Initially, GBS was primarily associated with the colonization of the mammary glands in cloven-hoofed animals, leading to bovine

**Peer Reviewer** Sebastian Cifuentes, Laboratorio de Referencia en Salud de Osorno, Valdivia, Chile

Address correspondence to Lina Guo, guo0201205@126.com, or Yingchun Xu, xycpumch@139.com.

Yingxing Li and Wenhang Yang contributed equally to this article. Author order was determined by contributions to this manuscript.

The authors declare no conflict of interest.

See the funding table on p. 15.

mastitis and impacting milk yield and quality (1, 2). In humans, GBS is an opportunistic pathogen that can intermittently, transiently, or persistently colonize the gastrointestinal and genitourinary tracts of healthy adults (3). Colonization rates vary by ethnicity, region, and age, ranging from 10 to 30% (4, 5). GBS is a leading cause of morbidity and mortality among infants in both high- and low-income countries (6). Recently, there has been growing clinical concern about invasive GBS infections in adults, particularly in individuals with underlying medical conditions or the elderly (7–9).

To evaluate the changing epidemiology of GBS burden and GBS antibiotic susceptibility in China over the past decade, we conducted a retrospective analysis of GBS data gathered from a major tertiary hospital at Beijing, spanning from 2013 to 2023. This comprehensive study investigated changes in vaginal GBS detection rates among pregnant and non-pregnant women, the proportion of GBS among vaginal pathogens, the clinical characteristics of invasive GBS disease, and shifts in antibiotic susceptibility profiles of GBS isolates from various sample sources. We discovered the increasing burden and the strikingly high multi-drug resistance rate of GBS in the past decade.

## MATERIALS AND METHODS

### Collection of data

This retrospective study was carried out at Peking Union Medical College Hospital (PUMCH) in Beijing, China, a class A tertiary hospital that holds the leading position in the National Tertiary Public Hospital Performance Evaluation in China. From January 2013 to December 2023, all patients with positive GBS cultures were included in this study. The method for calculating the GBS detection rate in pregnant and non-pregnant women involved dividing the number of positive GBS cultures from vaginal or rectovaginal swabs by the total number of vaginal or rectovaginal swabs collected each year. The criteria for defining invasive GBS disease include the isolation of GBS from normally sterile sites, such as blood, abdominal fluid, pleural fluid, joint fluid, bronchoalveolar lavage, drainage fluid, or cerebrospinal fluid. It also involves skin and soft tissue infections, where GBS is detected in abscesses, pus, and wound secretions, accompanied by local or systemic signs and symptoms of inflammation. The patient's gender, age, underlying conditions, clinical symptoms, and other medical data were retrieved from medical records. For the same patient, only the first cases of GBS isolated from the same site within 1 year were retained for analysis.

### Detection and identification of GBS

Samples collected through vaginal swabs were directly cultured on Columbia blood agar (Oxoid, Wesel, Germany), and candidate GBS colonies were subjected to matrix-assisted laser desorption/ionization time-of-flight mass spectrometry (bioMérieux, Marcy l'Etoile, France) for identification. The detection of GBS from rectovaginal swabs was divided into three stages. In the first stage (2015–2018), rectovaginal samples were cultured in selective Lim broth (with 10 µg/mL colistin and 15 µg/mL nalidixic acid) for enrichment, and then subcultured on blood agar plates. In the second stage (2019–2022), rectovaginal samples were cultured in Strep B carrot broth (Autobio Diagnostics, China) for enrichment in addition to CHROMagar StrepB (Autobio Diagnostics, China) and blood agar plates. In the third stage (2023), rectovaginal samples were also cultured in Strep B carrot broth for enrichment in addition to CHROMagar StrepB and blood agar plates. Even the negative Strep B carrot broth (no color change) was subcultured on blood agar for further confirmation.

### Antimicrobial susceptibility testing

GBS isolates were further characterized to determine the antimicrobial susceptibility profile. The antibiotic susceptibility of GBS to penicillin G, ceftriaxone, vancomycin, linezolid, erythromycin, clindamycin, chloramphenicol, and levofloxacin was assessed

using either the disk diffusion method or the broth microdilution method, following the CLSI M100 ED34 guidelines. The broth microdilution method was performed using the VITEK 2 AST-P639 card (bioMérieux, Marcy l'Etoile, France), an antibiotic susceptibility testing system specifically designed for Gram-positive bacteria. Both the disk diffusion and broth microdilution methods were employed to evaluate the susceptibility of GBS isolates to penicillin (susceptible: zone diameter ≥ 24 mm or MIC ≤ 0.12 μg/mL), vancomycin (susceptible: zone diameter ≥ 17 mm or MIC ≤ 1 μg/mL), linezolid (susceptible: zone diameter ≥ 21 mm or MIC ≤ 2 μg/mL), erythromycin (resistant: zone diameter ≤ 15 mm or MIC ≥ 1 μg/mL), and levofloxacin (resistant: zone diameter ≤ 13 mm or MIC ≥ 8 μg/mL). For ceftriaxone, clindamycin, and chloramphenicol, only the disk diffusion method was used, with susceptibility or resistance thresholds defined as follows: ceftriaxone (susceptible: zone diameter ≥ 24 mm), clindamycin (resistant: zone diameter ≤ 15 mm), and chloramphenicol (resistant: zone diameter ≤17 mm).

## Statistical analysis

The proportions of different groups were compared using Fisher's exact test with two-sided intervals. A *P* value less than 0.05 was considered significant. All analyses and graphics were performed by using GraphPad Prism (v8.0).

## RESULTS

### The distribution and trends of vaginal GBS carriage from 2013 to 2023

Screening for GBS colonization in pregnant women is crucial, as appropriate and timely antibiotic prophylaxis can significantly reduce early-onset GBS infection in infants. Initially, the vaginal swabs from pregnant women were directly cultured on blood agar plates to screen for vaginal GBS carriage in our hospital. However, this method has a high false negative rate. To improve the sensitivity of GBS screening in pregnant women, our hospital has continuously refined the screening methods in accordance with international guidelines (4, 10). These improvements have led to a significant increase in the detection rate of GBS in pregnant women.

In the past decade, GBS screening for pregnant women at our hospital has evolved through four distinct stages (Fig. 1A). The first stage (2013–2014) involved taking vaginal swabs from pregnant women and directly culturing them on blood agar plates. In the second stage (2015–2018), both vaginal and rectovaginal swabs were collected. Vaginal swabs were also directly cultured on blood agar plates, while rectovaginal swabs were enriched in Lim broth before being cultured on blood agar plates. The third stage (2019–2022) included sampling both vaginal and rectovaginal swabs, with the vaginal swabs directly cultured on blood agar plates and the rectovaginal swabs cultured on both blood agar plates, CHROMagar plates and Strep B carrot broth, for enrichment. In the fourth stage (2023), based on the third stage, an additional step was added, where if Strep B carrot broth showed no color change, the sample from broth was further cultured on blood agar plates for confirmation.

From 2015 to 2023, a total of 26,310 rectovaginal swabs from pregnant women at 35–37 weeks gestation were tested, with annual submissions ranging from 2,090 to 3,289 cases. As shown in Fig. 1B, improvements in GBS detection methods for rectovaginal swabs have led to a significant increase in the GBS detection rate. From 2015 to 2018, when rectovaginal swab sampling and Lim broth enrichment were first introduced, the average GBS detection rate for rectovaginal swabs in pregnant women (5.6%) was 2.5 times higher than the average GBS detection rate for vaginal swabs (2.2%) during the same period. From 2019 to 2022, after introducing Strep B carrot broth and CHROMagar plates, the average GBS detection rate for rectovaginal swabs (9.2%) increased by 65.1% compared to the second stage (5.6%). In 2023, with the introduction of the fourth-stage detection method for rectovaginal swabs, the GBS detection rate increased by another 15.4% compared to the 2019–2022 average. With these improvements, the GBS detection rate for rectovaginal swabs in 2023 was 10.6%, which should accurately reflect the GBS carriage rate in late-term pregnant women.

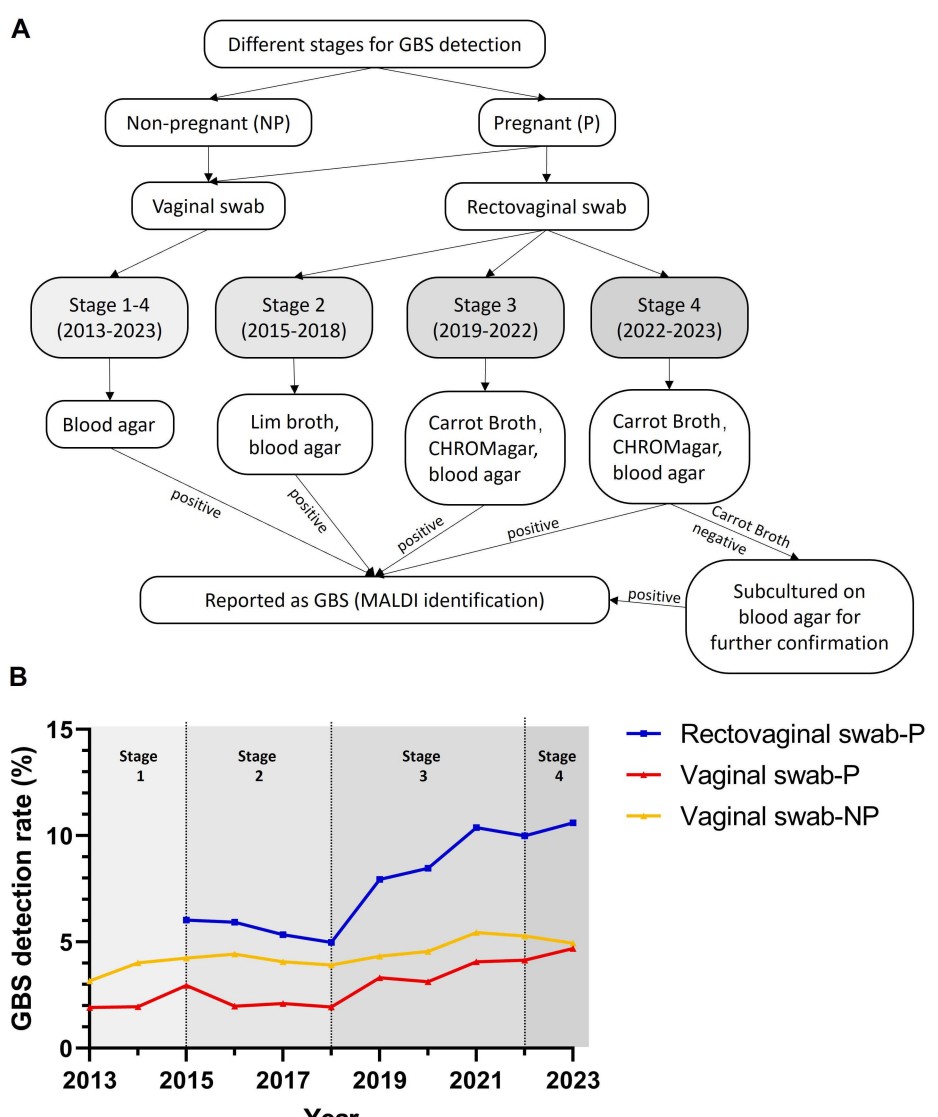

**FIG 1** (A) Evolution of the GBS detection method at PUMCH from 2013 to 2023. (B) Upper panel: the trends of the vaginal GBS detection rate from pregnant and non-pregnant women in the past decade. Lower panel: the sample size (N) of swabs submitted for each group per year. P indicates pregnant women. NP indicates non-pregnant women.

| Sample size (N) | Stage 1 | | Stage 2 | | | | Stage 3 | | | | Stage 4 |
|---|---|---|---|---|---|---|---|---|---|---|---|
| | 2013 | 2014 | 2015 | 2016 | 2017 | 2018 | 2019 | 2020 | 2021 | 2022 | 2023 |
| Rectovaginal swab-P | | | 2,090 | 2,968 | 2,788 | 2,912 | 3,036 | 2,785 | 3,102 | 3,233 | 3,298 |
| Vaginal swab-P | 3,337 | 3,629 | 2,739 | 3,579 | 3,238 | 3,392 | 3,432 | 3,325 | 3,467 | 3,548 | 3,795 |
| Vaginal swab-NP | 3,249 | 4,253 | 4,100 | 4,921 | 5,184 | 5,282 | 4,994 | 3,625 | 4,992 | 5,414 | 7,551 |

Since the GBS detection rate of rectovaginal swabs is influenced by improvements in detection methods, it may not accurately reflect the trend of GBS colonization in pregnant women. However, the GBS detection method for vaginal swabs in pregnant women has remained unchanged over the past 11 years, allowing us to infer the trend of GBS colonization. From 2013 to 2023, a total of 37,481 vaginal swabs from pregnant women were tested, with annual submissions ranging from 2,739 to 3,795 cases. As shown in Fig. 1B, since 2018, the GBS detection rate for vaginal swabs in pregnant

women has been slowly increasing, reaching 4.7% in 2023, depicting a significant increase of 145% compared to the GBS detection rate in 2013 (1.9%) ($P < 0.05$).

We also analyzed the non-pregnant women who underwent vaginal swab testing at our hospital over the past 11 years, totaling to 53,565 cases. Each year, the GBS detection rate for non-pregnant women was higher than that for pregnant women, with the 11-year average GBS detection rate for non-pregnant women (4.5%) being 1.5 times that of pregnant women (2.9%). This may be because most non-pregnant women undergoing vaginal swab testing at our hospital have certain related clinical symptoms. The GBS carriage rate in non-pregnant women has also slowly increased over the past 11 years, though not as markedly as in pregnant women. In 2023, the GBS detection rate for vaginal swabs in non-pregnant women was 4.9%, showing a 53% increase compared to the GBS detection rate in 2013 (3.2%) ($P < 0.05$).

To determine the distribution of vaginal GBS detection rates across different age groups, we categorized both pregnant and non-pregnant women by age. Pregnant women were divided into two groups: those aged 35 and older, and those under 35. Non-pregnant women were divided into four groups: minors (15 years old or younger), young adults (16–39 years old), middle-aged adults (40–64 years old), and seniors (65 years old or older). As shown in Fig. 2A, there was no significant difference in the GBS detection rates between the two age groups of pregnant women for both rectovaginal and vaginal swabs ($P > 0.05$). However, as shown in Fig. 2B, there were significant differences in the vaginal GBS detection rates among non-pregnant women of different ages. The detection rate was lowest in the minor group, significantly lower than the other three groups ($P < 0.05$). The middle-aged group had the highest detection rate, significantly higher than those of the other three age groups ($P < 0.05$). The detection rates in the young adult and senior groups were similar.

To assess the proportion of GBS among vaginal pathogens detected in pregnant and non-pregnant women, we analyzed the pathogen types in 6,418 vaginal swabs from pregnant women and 13,672 vaginal swabs from non-pregnant women that tested positive for microbial cultures between 2013 and 2023. In pregnant women, fungi

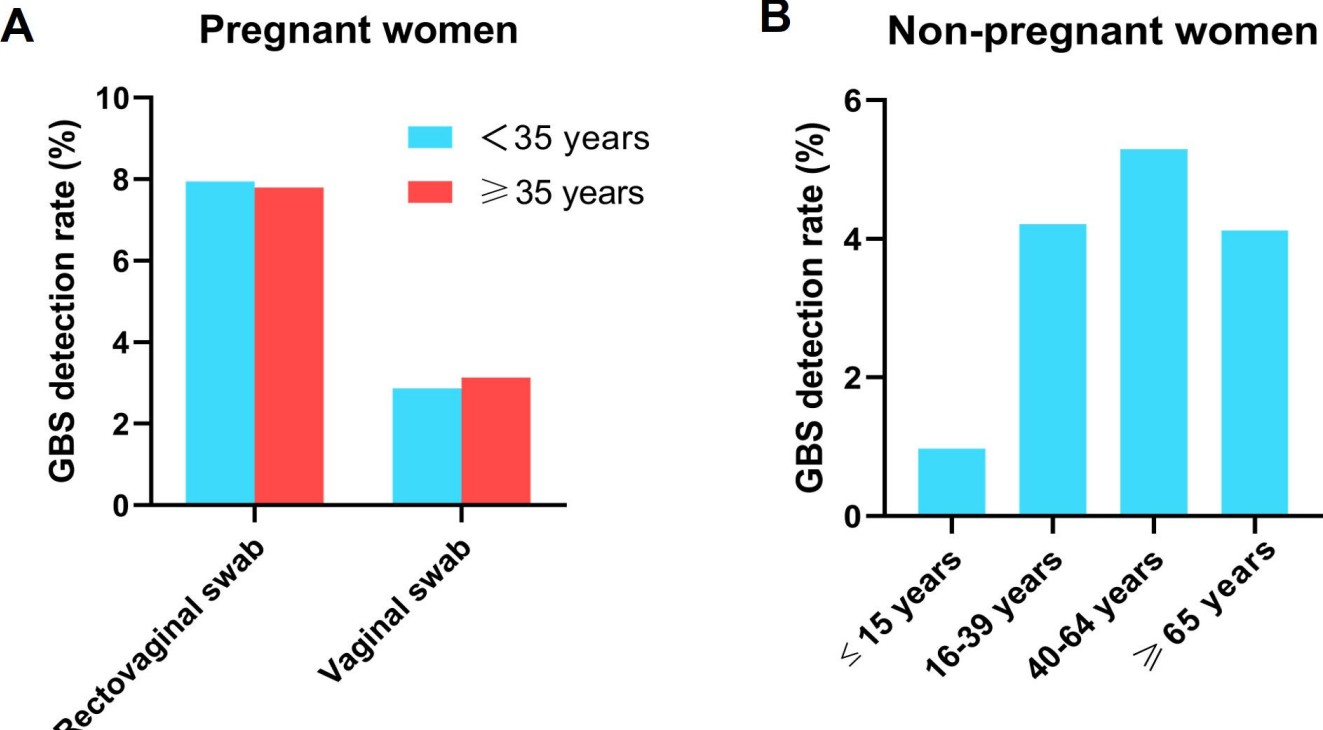

**FIG 2** Distribution of the vaginal GBS detection rates across different age groups in pregnant (A) and non-pregnant women (B).

were the most prevalent type of pathogens, accounting for an average of 59% over 11 years. Gram-negative bacteria were the least common, with an average proportion of 9%, while Gram-positive bacteria were in-between at 31.5%. Notably, the proportion of Gram-positive bacteria had significantly increased since 2021, reaching 42% in 2023, which is 45.8% higher than the average proportion before 2021 ($P < 0.05$) (Fig. 3A). The pathogen composition in vaginal swabs of non-pregnant women was significantly different from that of pregnant women. In non-pregnant women, the variations in the proportions of fungi, Gram-negative, and Gram-positive bacteria were not as pronounced. Gram-positive bacteria were the most prevalent (38.4%), followed by fungi (34.8%) and Gram-negative bacteria (25.7%). Compared to pregnant women, the proportion of fungi in non-pregnant women's vagina decreased, while that of Gram-negative bacteria increased (Fig. 3A).

To further clarify the proportion and changes of GBS among vaginal pathogens, we analyzed the proportion of different pathogens at the species level over 11 years in both pregnant and non-pregnant women. We selected the most abundant fungi species (*Candida albicans* and *Candida glabrata*), the most abundant Gram-positive

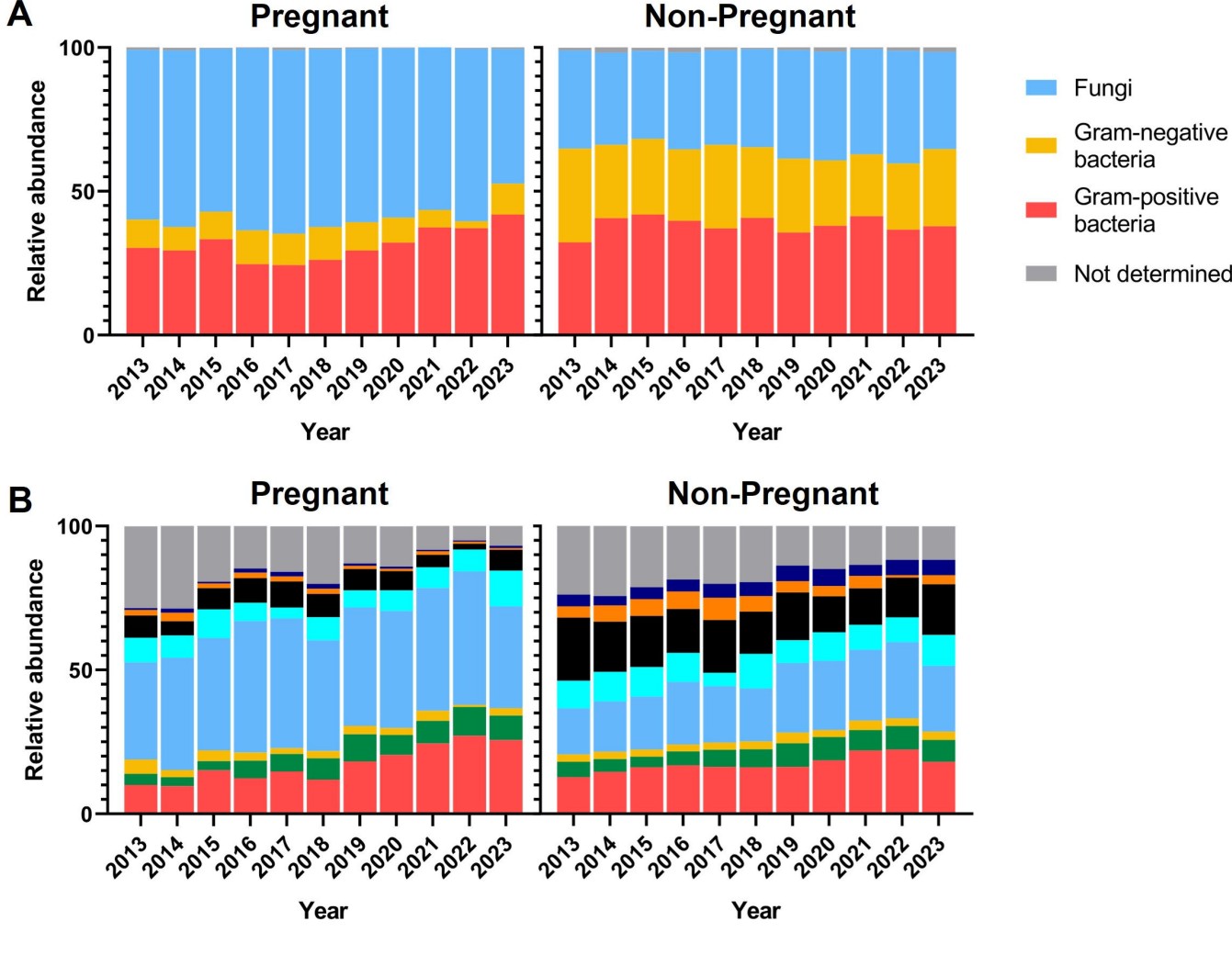

**FIG 3** (A) Composition of vaginal pathogens in pregnant and non-pregnant women from 2013 to 2023. (B) Proportions of predominant pathogen species of vaginal swabs in pregnant and non-pregnant women from 2013 to 2023.

bacteria (GBS, *Gardnerella vaginalis*, *Enterococcus faecalis*, and *Staphylococcus aureus*), and the most abundant Gram-negative bacteria (*Escherichia coli* and *Klebsiella pneumoniae*) detected in vaginal swabs for analysis. In the vaginal swabs of pregnant women, the pathogens were ranked by proportion as follows: *C. albicans* (40.7%), GBS (17.3%), *G. vaginalis* (7.7%), *E. coli* (6.7%), *C. glabrata* (6.6%), *S. aureus* (2.8%), *E. faecalis* (1.5%), and *K. pneumoniae* (1%). In non-pregnant women, the ranking was: *C. albicans* (21.3%), GBS (17.3%), *E. coli* (16.2%), *G. vaginalis* (9.3%), *C. glabrata* (6.3%), *K. pneumoniae* (4.7%), *E. faecalis* (4.6%), and *S. aureus* (2.8%) (Fig. 3B). Compared to pregnant women, the proportion of *C. albicans* significantly decreased in non-pregnant women, while those of *E. coli*, *E. faecalis*, and *K. pneumoniae* increased significantly. The proportion of *G. vaginalis* increased slightly, while those of GBS, *C. glabrata*, and *S. aureus* remained unchanged. Since the non-pregnant group mainly consisted of patients with related clinical symptoms, this suggested that *E. coli*, *E. faecalis*, and *K. pneumoniae* were more likely to be associated with pathogenic conditions. In contrast, GBS, *C. glabrata*, and *S. aureus* might be asymptomatic carriers, and *C. albicans* might be competitively inhibited by Gram-negative bacteria.

The trend of GBS proportion showed a gradual increase in both pregnant and non-pregnant groups (Fig. 3B). The annual number of vaginal swabs testing positive for pathogens, along with the proportion of GBS among vaginal pathogens (%), is presented in Fig. S1. In the pregnant group, the proportion of GBS reached 25.7% in 2023, which was 2.5 times higher than that in 2013 (10.1%) ($P < 0.05$). In the non-pregnant group, the proportion of GBS reached 18.1% in 2023, showing a 41.4% increase compared to that in 2013 (12.8%) ($P < 0.05$). The increasing proportion of GBS in the vaginal pathogens correlates with the rising GBS detection rate in vaginal swabs of pregnant women, highlighting the significant presence of GBS in pregnant women and the increasing trend of vaginal GBS colonization. This further emphasizes the importance of GBS screening and prophylactic treatment for pregnant women in China.

Next, we analyzed the composition of vaginal pathogens detected in different age groups of pregnant and non-pregnant women. We found that in the pregnant group, the vaginal pathogen composition was very similar between those aged 35 and above and those under 35 (Fig. 4A). The top three pathogen species were *C. albicans*, GBS, and *G. vaginalis*. However, in the non-pregnant group, there were significant differences in the vaginal pathogen composition across different age groups (Fig. 4A). In the group of minors aged 15 and under, the top three most abundant vaginal pathogens were *E. coli* (21.4%), *C. albicans* (20.3%), and *S. aureus* (8.7%), with GBS accounting for only 2.9%. In the young adult group (16–39 years old), the top three were *C. albicans* (25.8%), GBS (17.9%), and *E. coli* (13.2%). In the middle-aged adult group (40–64 years old), the

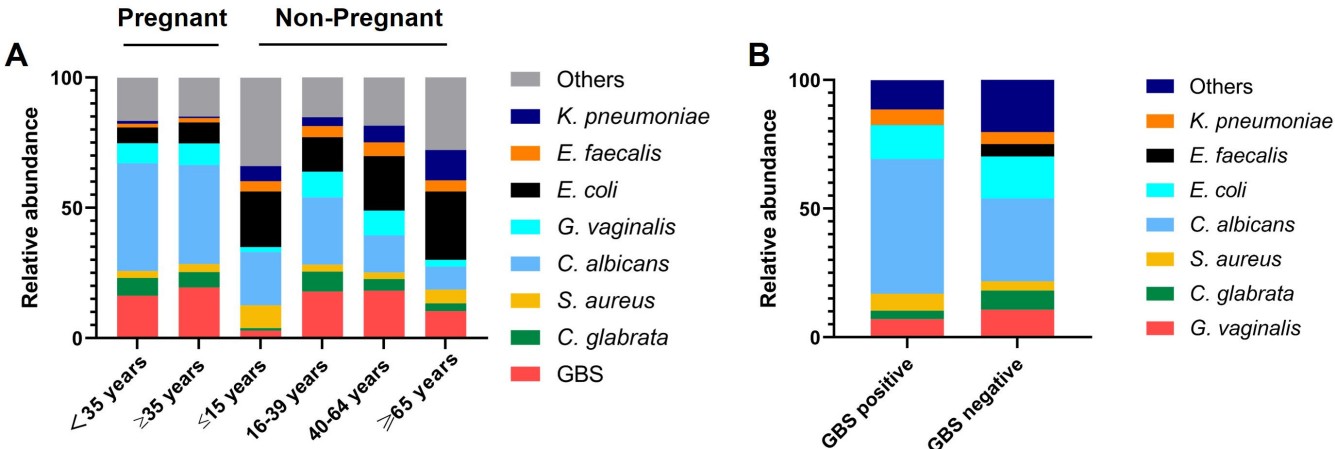

**FIG 4**  (A) Vaginal pathogen composition of pregnant and non-pregnant women across different age groups. (B) Vaginal pathogen composition in GBS positive group or GBS negative group.

top three were *E. coli* (20.9%), GBS (18.2%), and *C. albicans* (14.2%). In the elderly group (65 years and older), the top three were *E. coli* (26.3%), *K. pneumoniae* (11.7%), and GBS (10.4%). From this, we observed that in the vaginal pathogens of non-pregnant adults, the proportion of *C. albicans* decreased with age, while that of *E. coli* increased with age. The proportion of GBS was higher in the 16–64 age group compared to minors and the elderly. In all age groups, GBS remained an important component of the adult vaginal pathogens.

To explore potential mutualistic or competitive relationships between GBS and other vaginal pathogens, we categorized vaginal swabs with microbial growth into those containing GBS and those without GBS and analyzed the differences in pathogen composition between the two groups (Fig. 4B). The most significant change was observed in the proportion of *E. faecalis*, which was 4.8% in swabs without GBS, but 0% in swabs with GBS, indicating a strong competitive or antagonistic relationship between GBS and *E. faecalis* in the vaginal environment. Meanwhile, *C. albicans* and GBS might have a certain mutualistic relationship, as the proportion of *C. albicans* in swabs without GBS was 32.2%, while it increased to 52.3% in swabs with GBS (Fig. 4B).

## The clinical characteristics of invasive GBS diseases from 2013 to 2023

A total of 165 patients were identified as invasive GBS (iGBS) infection during the 11-year study period. Among them were three neonatal cases, one with early-onset disease and the other two with late-onset disease. GBS was detected in the peripheral blood of all three neonates and in the cerebrospinal fluid of one of the late-onset cases. The remaining 162 iGBS cases were adults aged over 16, with only one pregnant woman among them. The number of iGBS cases categorized by age group for each year from 2013 to 2023 seemed to have risen since 2016, especially among middle-aged and elderly patients aged 40 and above (Fig. 5). However, there was not an apparent increase in the number of cases among patients aged 16–39.

Table 1 presents the basic clinical information for these 162 adult iGBS cases. Out of the 162 cases, 89 patients were female (54.9%), a proportion that remained stable across different age groups. A total of 134 patients had underlying diseases (82.7%), with varying prevalence across age groups—62.8% of patients aged 16–39 had underlying diseases, while the percentage was 93.8% for patients aged 65 and above. Many patients with iGBS infection had more than one underlying condition. In general, the six most

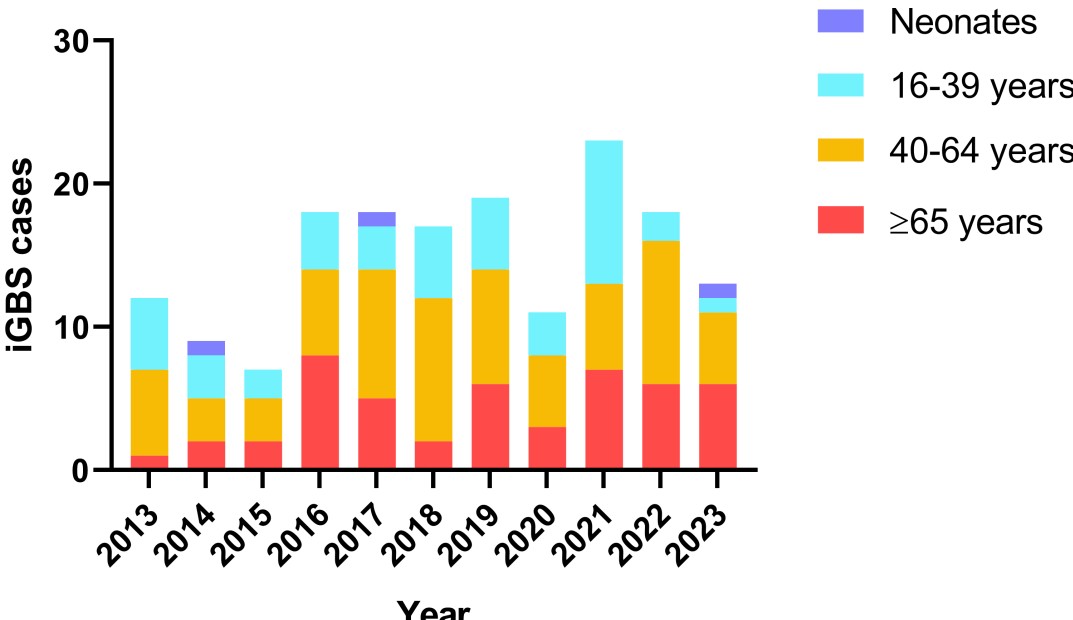

**FIG 5** Number of iGBS cases by age group for each year from 2013 to 2023.

common underlying conditions were cancer (39.5%), cardiovascular disease (34.0%), diabetes (27.2%), autoimmune diseases (13.6%), liver disease (10.5%), and peripheral vascular disease (8.6%). However, the proportions and types of common underlying diseases vary slightly among different age groups. For instance, the prevalence of cardiovascular disease, diabetes, and peripheral vascular disease is significantly higher in elderly patients over 65 years old compared to younger patients. Among patients aged 16–39, the most common types of underlying conditions were cancer (25.6%), autoimmune diseases (16.3%), and liver disease (16.3%). In middle-aged patients aged 40–64, the most common types of underlying diseases were cancer (46.5%), cardiovascular disease (28.1%), and diabetes (26.8%). Among elderly patients aged 65 and older, the most common types of underlying diseases were cardiovascular disease (66.7%), diabetes (43.8%), and cancer (41.7%).

The three most common clinical manifestations of iGBS infection were skin and soft tissue infection (42.0%), bacteremia without focus (22.8%), and intra-abdominal infection (12.3%). The distribution of these clinical manifestations did not vary significantly across all age groups. Other common manifestations included pneumonia (7.4%), abscess (4.3%), septic shock-associated infection (4.3%), invasive urinary tract infection (4.3%), and osteomyelitis (2.5%). Notably, septic shock-associated infection was only present in middle-aged and elderly patients aged 40 and older, indicating that younger patients were less likely to progress to severe conditions. Among the 162 cases of iGBS infection in adults, 43.2% involved polymicrobial infections. However, the proportion of polymicrobial infections varied greatly among different types of infections. The clinical manifestations with the highest proportions of polymicrobial infections were skin and soft tissue infections (69.1%), intra-abdominal infections (55%), and abscesses (42.9%).

**TABLE 1** Characteristics of adults with invasive GBS infections per age category

| Characteristics | Cases, by age group | | | | P value |
|---|---|---|---|---|---|
| | All | 16–39 years | 40–64 years | ≥65 years | |
| | (n = 162) | (n = 43) | (n = 71) | (n = 48) | |
| Female gender | 89 (54.9%) | 23 (53.5%) | 39 (54.9%) | 27 (56.3%) | 0.979 |
| Underlying condition | | | | | |
| ≥1 condition | 134 (82.7%) | 27 (62.8%) | 62 (87.3%) | 45 (93.8%) | <0.001 |
| Cancer | 64 (39.5%) | 11 (25.6%) | 33 (46.5%) | 20 (41.7%) | 0.076 |
| Cardiovascular disease | 55 (34.0%) | 3 (7.0%) | 20 (28.1%) | 32 (66.7%) | <0.001 |
| Diabetes mellitus | 44 (27.2%) | 4 (9.3%) | 19 (26.8%) | 21 (43.8%) | <0.001 |
| Autoimmune disease | 22 (13.6%) | 7 (16.3%) | 10 (14.1%) | 5 (10.4%) | 0.736 |
| Liver disease | 17 (10.5%) | 7 (16.3%) | 6 (8.5%) | 4 (8.3%) | 0.37 |
| Peripheral vascular disease | 14 (8.6%) | 0 (0.0%) | 7 (9.9%) | 7 (14.6%) | 0.024 |
| Renal disease | 9 (5.6%) | 1 (2.3%) | 6 (8.5%) | 2 (4.2%) | 0.422 |
| Neurologic disorders | 9 (5.6%) | 0 (0.0%) | 4 (5.6%) | 5 (10.4%) | 0.097 |
| Arthritis | 7 (4.3%) | 1 (2.3%) | 1 (1.4%) | 5 (10.4%) | 0.051 |
| Lung disease | 6 (3.7%) | 1 (2.3%) | 4 (5.6%) | 1 (2.1%) | 0.66 |
| Others | 11 (6.8%) | 4 (9.3%) | 3 (4.2%) | 1 (2.1%) | 0.295 |
| Clinical manifestations | | | | | |
| Skin and/or soft tissue infection | 68 (42.0%) | 20 (46.5%) | 27 (38%) | 21 (43.8%) | 0.65 |
| Bacteremia without focus | 37 (22.8%) | 12 (27.9%) | 13 (18.3%) | 12 (25%) | 0.425 |
| Intra-abdominal infection | 20 (12.3%) | 3 (7.0%) | 13 (18.3%) | 4 (8.3%) | 0.158 |
| Pneumonia | 12 (7.4%) | 1 (2.3%) | 7 (9.9%) | 4 (8.3%) | 0.376 |
| Abscess | 7 (4.3%) | 3 (7.0%) | 2 (2.8%) | 2 (4.2%) | 0.551 |
| Septic shock-associated | 7 (4.3%) | 0 (0.0%) | 5 (7%) | 2 (4.2%) | 0.231 |
| Invasive urinary tract infection | 7 (4.3%) | 1 (2.3%) | 2 (2.8%) | 4 (8.3%) | 0.364 |
| Osteomyelitis | 4 (2.5%) | 1 (2.3%) | 1 (1.4%) | 2 (4.2%) | 0.814 |
| Others | 7 (4.3%) | 2 (4.7%) | 4 (5.6%) | 1 (2.1%) | 0.71 |
| Polymicrobial infections | 70 (43.2%) | 18 (41.9%) | 31 (43.7%) | 21 (43.8%) | 1 |

**TABLE 2** Antibiotic resistance rates of GBS isolates categorized by specimen types

| Resistance rate | Erythromycin | Chloramphenicol | Clindamycin | Levofloxacin |
|---|---|---|---|---|
| Total | 72.2% (3,698/5,121) | 9.5% (379/3,982) | 60.0% (2,431/4,050) | 50.1% (2,910/5,810) |
| Vagina | 73.7% (2,245/3,046) | 10.6% (2,62/2,471) | 60.7% (1,517/2,501) | 50.4% (1,543/3,064) |
| Rectovaginal swab | 70.0% (1,142/1,637) | 7.7% (96/1,246) | 57.5% (725/1,261) | 43.6% (718/1,648) |
| Urine | 64.2% (52/81) | 5.9% (4/68) | 54.9% (39/71) | 60.5% (446/737) |
| Sputum | 69.4% (34/49) | 14.7% (5/34) | 68.4% (26/38) | 54.9% (28/51) |
| Blood | 70.2% (33/47) | 11.1% (2/18) | 58.3% (14/24) | 66.7% (32/48) |
| Drainage | 65.7% (23/35) | 0% (0/19) | 63.2% (12/19) | 65.7% (23/35) |
| Wound | 74.3% (26/35) | 13.6% (3/22) | 77.3% (17/22) | 57.1% (20/35) |
| Pus | 79.3% (23/29) | 15.4% (2/13) | 73.3% (11/15) | 48.3% (14/29) |
| Secretion | 81.5% (22/27) | 5.6% (1/18) | 75% (15/20) | 64.3% (18/28) |
| Skin | 66.7% (14/21) | 0% (0/10) | 63.6% (7/11) | 47.6% (10/21) |
| Semen | 73.3% (11/15) | 0% (0/9) | 33.3% (3/9) | 26.7% (4/15) |
| Rectal (neonates) | 66.7% (12/18) | 0% (0/9) | 66.7% (6/9) | 43.8% (7/16) |
| Throat | 83.3% (10/12) | 18.2% (2/11) | 90.9% (10/11) | 66.7% (8/12) |
| Tissue | 91.7% (11/12) | 0% (0/4) | 50% (2/4) | 41.7% (5/12) |
| Mouth (neonates) | 54.5% (6/11) | 0% (0/2) | 66.7% (2/3) | 58.3% (7/12) |
| Tracheobronchial aspirate | 100% (8/8) | 0% (0/3) | 100% (4/4) | 50% (4/8) |
| Pleural fluid | 60% (3/5) | 0% (0/4) | 50% (2/4) | 66.7% (4/6) |
| Abdominal fluid | 60.0% (3/5) | 0% (0/2) | 100.0% (3/3) | 60.0% (3/5) |
| Broncho-alveolar lavage | 100% (4/4) | 0% (0/2) | 100% (2/2) | 100% (4/4) |
| Catheter | 100% (2/2) | 0% (0/1) | 100% (1/1) | 100% (2/2) |
| Joint fluid | 100% (2/2) | 0% (0/1) | 100% (1/1) | 100% (2/2) |
| Breast milk | 50% (1/2) | 0% (0/2) | 100% (2/2) | 50% (1/2) |
| Ear | 50% (1/2) | 0% (0/1) | 100% (1/1) | 0% (0/2) |
| Umbilical cord (neonates) | 50% (1/2) | 0% (0/2) | 50% (1/2) | 50% (1/2) |
| Not determined | 58.3% (7/12) | 25% (2/8) | 70% (7/10) | 41.7% (5/12) |

The manifestations with the lowest proportions were bacteremia without a focus (7.7%), invasive urinary tract infections (14.3%), and septic shock-associated infections (14.3%).

## Shifts in GBS antibiotic susceptibility profiles from 2013 to 2023

Over the 11-year study period, a total of 5,858 unrelated GBS isolates were collected and examined for antibiotic susceptibility. Among these isolates, the antibiotic sensitivity of penicillin G (5,729 isolates), ceftriaxone (4,290 isolates), vancomycin (5,769 isolates), linezolid (5,831 isolates), erythromycin (5,125 isolates), chloramphenicol (3,982 isolates), clindamycin (4,050 isolates), and levofloxacin (5,810 isolates) were predominantly examined. All GBS isolates in our study remained susceptible to penicillin G, ceftriaxone, vancomycin, and linezolid, but exhibited a strikingly high resistance to erythromycin, clindamycin, and levofloxacin. The total resistance rate to erythromycin over 11 years was 72.2%, to clindamycin was 60.0%, and to levofloxacin was 50.1%, while the total resistance rate to chloramphenicol was relatively low at 9.5%. Additionally, the 5,858 GBS isolates exhibited a high multidrug-resistance rate of 30.8% against erythromycin, clindamycin, and levofloxacin concurrently.

Figure 6A shows the annual trends in resistance rates of GBS isolates collected in this study to erythromycin, chloramphenicol, clindamycin, and levofloxacin. The resistance rate of GBS isolates to erythromycin was slowly increasing. In 2013, the erythromycin resistance rate was only 62.9%, but by 2023, it had risen to 78.0%, marking a 24% increase compared to 2013 ($P < 0.05$). While the resistance rates to chloramphenicol, clindamycin, and levofloxacin fluctuated over time, no significant overall trend was observed.

All the 5,858 GBS isolates can be classified into 165 isolates causing invasive infections and 5693 isolates associated with non-invasive infections or colonization. There was no significant difference in the resistance rates to erythromycin, chloramphenicol, and

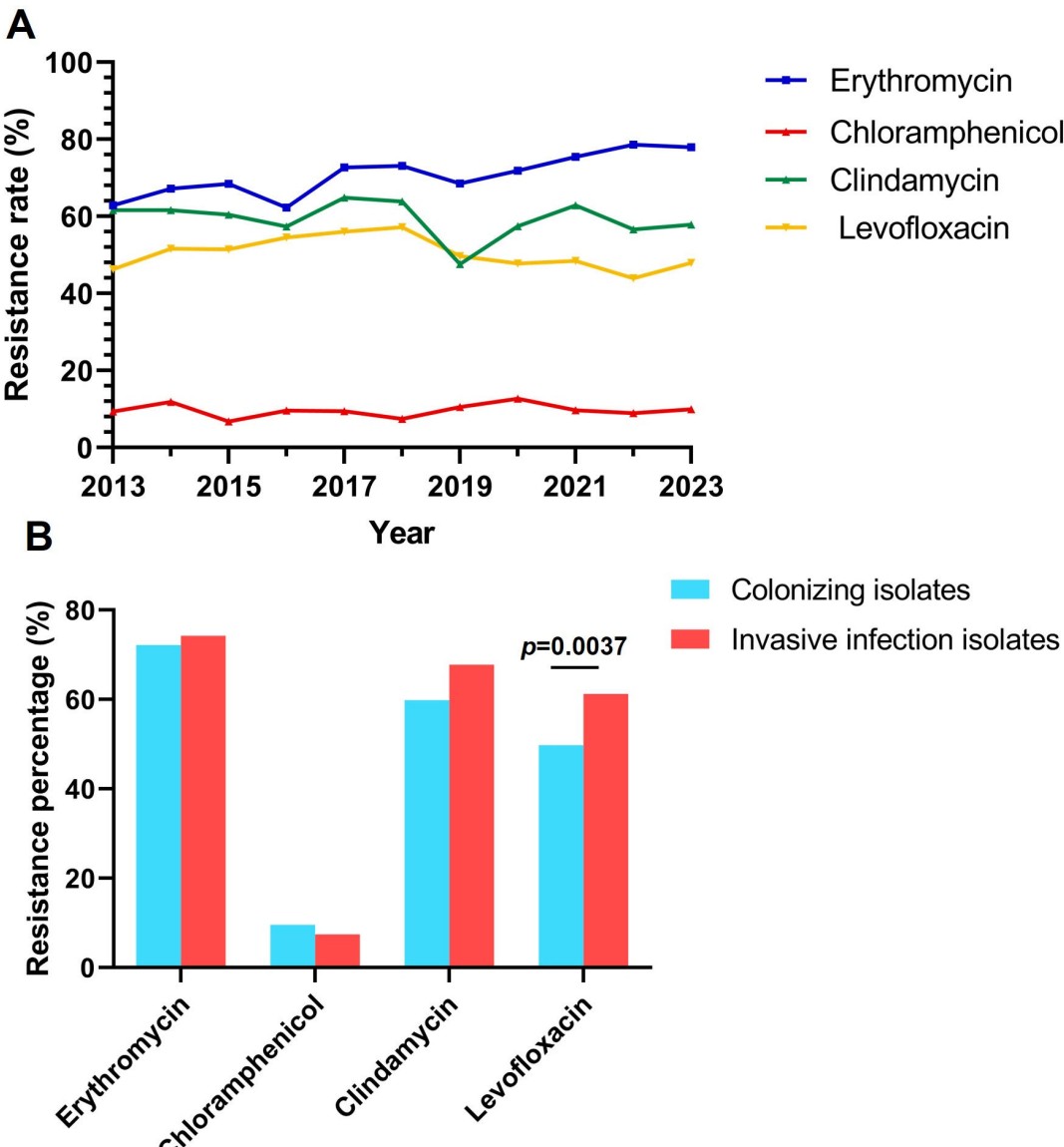

**FIG 6** (A) Antibiotic resistance rates of GBS for each year from 2013 to 2023. (B) Difference in antibiotic resistance rate between GBS colonizing isolates and GBS invasive infection isolates.

clindamycin between these two different types of isolates. However, the isolates causing invasive infections exhibited a significantly higher resistance rate to levofloxacin (61.2%) compared to the resistance rate of colonizing isolates (49.8%) (Fig. 6B).

Further analysis of specimen types with more than 10 cases (Table 2) reveals the following antibiotic resistance patterns: erythromycin resistance was higher in tissue (91.7%), throat (83.3%), and secretion (81.5%) samples; chloramphenicol resistance was elevated in throat (18.2%), pus (15.4%), and wound (13.6%) samples; clindamycin resistance was higher in throat (90.9%), wound (77.3%), and secretion (75%) samples; and levofloxacin resistance was more prevalent in blood (66.7%), throat (66.7%), and drainage (65.7%) samples. Notably, GBS isolates from throat samples exhibited high resistance levels across all four antibiotics, whereas those from rectovaginal swabs showed resistance rates lower than the overall average. As previously mentioned, GBS isolates responsible for invasive infections displayed significantly higher levofloxacin resistance than colonizing strains. Further examination of levofloxacin resistance across sample types revealed that isolates from sterile sites, such as blood, drainage, pleural

**TABLE 3** Antibiotic resistance rates of invasive GBS isolates categorized by representative clinical characteristics of patients

| Resistant rate | Erythromycin | Chloramphenicol | Clindamycin | Levofloxacin |
| --- | --- | --- | --- | --- |
| **Total** | **75.2% (121/161)** | **8.9% (7/79)** | **69.3% (61/88)** | **61.3% (100/163)** |
| Age group | | | | |
| Neonates | 100% (3/3) | 0% (0/2) | 66.7% (2/3) | 66.7% (2/3) |
| 16–39 years | 67.5% (27/40) | 15.8% (3/19) | 71.4% (15/21) | 55% (22/40) |
| 40–64 years | 80% (56/70) | 8.8% (3/34) | 73.7% (28/38) | 59.2% (42/71) |
| ≥65 years | 72.9% (35/48) | 4.2% (1/24) | 61.5% (16/26) | 69.4% (34/49) |
| Underlying condition | | | | |
| ≥1 condition | 74.2% (92/124) | 6.7% (4/60) | 71.2% (47/66) | 65.1% (82/126) |
| Cancer | 77.8% (49/63) | 6.7% (2/30) | 77.4% (24/31) | 62.5% (40/64) |
| Cardiovascular disease | 71.4% (40/56) | 4.3% (1/23) | 65.4% (17/26) | 66.1% (37/56) |
| Diabetes mellitus | 65.1% (28/43) | 5.3% (1/19) | 59.1% (13/22) | 65.9% (29/44) |
| Autoimmune disease | 87.0% (20/23) | 8.3% (1/12) | 100% (13/13) | 60.9% (14/23) |
| Liver disease | 77.8% (14/18) | 20.0% (1/5) | 83.3% (5/6) | 66.7% (12/18) |
| Peripheral vascular disease | 71.4% (10/14) | 16.7% (1/6) | 100% (6/6) | 64.3% (9/14) |
| Clinical manifestations | | | | |
| Skin or soft tissue infection | 75.4% (49/65) | 14.3% (5/35) | 73.7% (28/38) | 55.4% (36/65) |
| Bacteremia without focus | 73.7% (28/38) | 13.3% (2/15) | 73.7% (14/19) | 69.2% (27/39) |
| Intra-abdominal infection | 73.7% (14/19) | 0% (0/9) | 60% (6/10) | 63.2% (12/19) |
| Pneumonia | 72.7% (8/11) | 0% (0/8) | 62.5% (5/8) | 66.7% (8/12) |

fluid, abdominal fluid, and joint fluid, exhibited resistance rates exceeding 60%, markedly higher than the overall average resistance rate (50.1%).

The antibiotic susceptibility profile of GBS isolates from patients with invasive infections is presented in Table 3, categorized by age groups, representative underlying conditions (with more than 10 cases), and clinical manifestations (with more than 10 cases). Several notable patterns emerged from the data. For instance, the resistance rate of GBS to levofloxacin increased with patient age in cases of invasive infections. Additionally, the lowest levofloxacin resistance rate was observed in skin or soft tissue infections (55.4%), while the highest resistance rate was found in cases of bacteremia without a defined focus (69.2%).

## DISCUSSION

To increase the detection sensitivity of GBS carriage for pregnant women, our hospital started using rectovaginal swab sampling since 2015. From 2015 to 2018 (Stage 2), rectovaginal swabs were placed in Lim broth for enrichment after sampling and plated. From 2019 to 2022 (Stage 3), rectovaginal swabs were cultured on both blood agar plates, CHROMagar plates and Strep B carrot broth, for enrichment. In 2023 (Stage 4), samples from Strep B carrot broth were further subcultured on blood agar plates for confirmation, even if the broth showed no color change. Meanwhile, vaginal swabs continued to be directly plated on blood agar from 2013 to 2023. Since the gastrointestinal tract is a colonization site for GBS in the human body, many studies have found a higher detection rate of GBS in rectovaginal swabs compared to vaginal swabs (11–13). In our study, the average GBS detection rate in rectovaginal swabs was 2.5 times higher than that in vaginal swabs, slightly higher than the conclusion of the study by Aila et al., who found that the detection rate of GBS in rectovaginal swabs was twice than that in vaginal swabs. This difference in detection rates may be attributed to variations in post-sampling processes beyond the sampling location, which can also impact the sensitivity of GBS detection. In addition to the influence of sampling site, Lim broth enrichment, Strep B carrot broth enrichment, and CHROMagar plates, compared to direct plating on blood agar, also contribute to enhancing the sensitivity of GBS detection (10, 14, 15). Literature reports that an enrichment broth culture, in addition to direct plating, detected a further 4% of GBS carriers (16). However, some studies suggest that the increase in sensitivity for GBS detection with broth enrichment culture compared

to direct plating is very limited (17, 18). Taking all this information into account, we speculated that the significant increase in the detection rate of GBS in rectovaginal swabs compared to vaginal swabs primarily stemmed from the changes in sampling sites. Furthermore, to obtain GBS screening results more quickly, we also tested a non-culture-based PCR method. However, we found that its positivity rate was only one-third that of vaginal swabs (data not shown), consistent with the CDC's recommendation against using PCR as a routine screening method for GBS during delivery (19).

Our data showed that from 2013 to 2023, the prevalence of GBS carriage in pregnant women has been gradually increasing. Globally, the average GBS colonization rate among pregnant women is 18%, varying between 10 and 30% depending on region and ethnicity (4, 5). Pregnant women in East Asia and South Asia generally exhibit the lowest GBS colonization rates at around 10%. Specifically, the GBS carriage rates are 8% in the Republic of Korea, 10% in India, 11% in China, and relatively higher at 16% in Japan (5). In our data from 2015 to 2018, the GBS detection rates in rectovaginal swabs from pregnant women were only 5% to 6%, lower than the previously reported average GBS carriage rate among pregnant women in China. This discrepancy is likely due to regional variations in GBS colonization rates among pregnant women across China. For example, the lowest colonization rates are in Shanghai at 3.7% and Jiangsu at 4.2%, while the highest rates are in Fujian at 14.5% and Guangdong at 14.1% (20). In a study from 2014, the GBS carriage rate among 2,850 pregnant women in Beijing was reported to be 7.1% (21), which is similar to our findings. However, even among pregnant women in Beijing, the GBS carriage rate has shown a significant increase starting from 2019, reaching 10.6% by 2023. Notably, the proportion of GBS in vaginal pathogens detected in pregnant women has steadily increased from 10.1 to 25.7% from 2013 to 2023. This trend indicates a growing competitive advantage of GBS over other pathogens in the vaginal, highlighting the continued importance of GBS screening in pregnant women.

It was reported that the GBS colonization rate in healthy elderly women was similar to that in pregnant women (22). However, in our results, the average GBS detection rate in vaginal swabs of non-pregnant women (4.5%) was significantly higher than that of pregnant women (2.9%). Even in the non-pregnant group aged 65 and above, the GBS detection rate (4.1%) was significantly higher than in pregnant women. The possible reason for this discrepancy is that our sampled non-pregnant population mainly consisted of patients visiting the hospital, most of whom had related clinical symptoms. The top three clinical symptoms were vaginosis or vaginitis, infertility, and abdominal pain in our sampled non-pregnant population. Therefore, the types and proportions of vaginal pathogens in our non-pregnant population might not represent those in a healthy population. Additionally, our detection methods targeted only common pathogenic microorganisms and did not include bacteria, such as *Lactobacillus* spp., which dominate the vaginas of healthy individuals (23, 24). We found that the proportion of *Candida albicans* among vaginal pathogens was significantly higher in pregnant women compared to non-pregnant women. This aligns with previous findings that *Candida* spp. is more common in the vaginas of healthy women dominated by *Lactobacillus* and less common in those with bacterial vaginosis characterized by dysbiosis (25). This consistency helps to validate the credibility of our results.

The potential relationship between changes in vaginal GBS carriage and the coronavirus disease 2019 (COVID-19) pandemic is also a noteworthy issue. We found that the vaginal GBS carriage rates may have been differently affected by the COVID-19 pandemic (2020–2022) in pregnant and non-pregnant groups. For pregnant women, the GBS carriage rate had already significantly increased before the onset of the pandemic in 2019, and this upward trend continued through 2023, after the pandemic ended. In fact, the GBS carriage rate in 2023 exceeded that of the pandemic years (2020–2022). Therefore, we speculate that the increase in vaginal GBS carriage rates in pregnant women is unlikely to be directly caused by the COVID-19 pandemic. However, in the non-pregnant group, there was no significant rise in GBS carriage rates before the pandemic in 2020. During the three pandemic years (2020–2022), the GBS carriage

rate in non-pregnant women significantly increased, but after the pandemic ended in 2023, it slightly declined, though it remained higher than pre-pandemic levels. Thus, we hypothesize that the COVID-19 pandemic may have contributed to the increase in GBS carriage in non-pregnant women. Nevertheless, more data in the coming years will be needed to infer the precise impact of the pandemic on GBS carriage rates.

During the 11-year period from 2013 to 2023, we had a total of 165 cases of GBS invasive infections, with only three cases from neonates (1.8%) and one case from pregnant women (0.62%). The lower infection rates in pregnant women and newborns may be attributed to the lower prevalence of maternal GBS colonization in Beijing and the appropriate antibiotic prophylaxis administered at our hospital. Our study validates the importance of GBS as a pathogen across all age groups, in line with previous research findings (2, 9, 26, 27). Among our 162 adult iGBS patients, 82.7% had at least one underlying disease. The most prevalent types were cancer, cardiovascular disease, and diabetes, aligning with previous literature reports (7, 9, 28). However, unlike previous reports (9, 28, 29), our data showed that iGBS patients aged 65 and above were not the most prevalent (29.1%), with iGBS patients aged 40–64 representing the largest group (43%). The proportion of polymicrobial infections in iGBS cases was 43.2%, higher than that reported in Latin America and Thailand (7, 30), possibly due to the higher proportion of skin and soft tissue infections in our iGBS cases. A study from the United States found that polymicrobial infections accounted for 40% of GBS infection cases, which is similar to our findings, with skin and soft tissue infections being the most common type of infection (31). Additionally, it is worth noting the low mortality rate demonstrated by our iGBS patients (1.8%), which was significantly lower compared to other literature reports (~10%) (7, 28, 29). Out of the 165 patients, three patients died, one of whom was a neonate with early-onset sepsis, while the other two patients were from the 40–64 age group and presented with abdominal infection and septic shock, respectively.

Although studies have reported rare GBS isolates with reduced penicillin susceptibility or vancomycin resistance (32), in our collection of 5,858 GBS isolates from the past 11 years, we did not find any isolates resistant to penicillin G, ceftriaxone, vancomycin, and linezolid, which is consistent with most previous studies (21, 33, 34). In our study, erythromycin resistance is particularly high at 72.2%, which is similar to the resistance rate previously reported in China but higher than that in other regions worldwide (35, 36). Furthermore, our data showed that over the past 11 years, the erythromycin resistance rate of GBS had been on the rise, reaching as high as 78.0% in 2023. The clindamycin resistance rate in our GBS isolates is also notably high on a global scale at 60% compared to Algeria (43.2%), Portugal (34%), Brazil (2%), and Ghana (3.1%) (37–40).

The high resistance rate of our GBS isolates to levofloxacin (50.1%) is noteworthy. Globally, GBS resistance to fluoroquinolones (FQ) is on a significant rise, particularly in East Asia, where isolates from Japan exhibit a resistance rate of 43.6% and from South Korea, 37.2% (41, 42). In certain European countries, such as France (1.5%) and Italy (2.99%) (33, 43), GBS resistance to FQ remains relatively low. In China, the resistance rate of GBS to levofloxacin also varies considerably by region, with the highest rates observed in Northeast China (67.7%) and the lowest in South China (24.0%) (44). Unusually, in the GBS samples we collected, the levofloxacin resistance rate among isolates causing invasive infections was significantly higher than that among colonizing isolates, inconsistent with a previous study that has reported higher FQ resistance rates among carriage isolates than infectious strains (33). Further analysis by specimen type revealed that GBS strains isolated from blood exhibited the highest levofloxacin resistance rate at 66.7%. Our findings suggest that the extensive use of FQ in Beijing, China could lead to the emergence of epidemic FQ-resistant GBS clones causing invasive infections.

In addition to high resistance rates to erythromycin, clindamycin, and levofloxacin, our GBS isolates also exhibited a high multidrug-resistance rate of 30.8% against these three antibiotics concurrently. This suggests that for patients allergic to β-lactam antibiotics, caution should be exercised in the empirical use of conventional second-line GBS treatment drugs, such as erythromycin, clindamycin, and levofloxacin. Conducting

susceptibility testing prior to initiating prophylaxis with these antibiotics is essential. Given the worrisome situation regarding GBS resistance, ongoing surveillance of GBS resistance rates is essential for the future.

## Conclusion

Our study shows that from 2013 to 2023, the detection rate of vaginal GBS in both pregnant and non-pregnant women in Beijing, China has been on the rise, with GBS increasingly comprising a larger proportion of vaginal pathogens. This possible competitive advantage of GBS among vaginal pathogens underscores the importance of GBS detection and screening. Among non-pregnant women, the detection rate of GBS and its proportion in the vaginal pathogens vary across different age groups. In patients with invasive GBS infections, 82% had underlying conditions, with cancer, cardiovascular disease, and diabetes being the most common. The GBS isolates we collected exhibited alarming resistance rates to erythromycin, clindamycin, and levofloxacin, with a high proportion of multidrug-resistant strains resistant to all three antibiotics. Particularly noteworthy is the significantly increased resistance to levofloxacin in invasive GBS, especially GBS isolated from blood samples, compared to colonizing GBS, which warrants our attention.

## ACKNOWLEDGMENTS

This work was financially supported by the National Key Research Program of China (2021YFC2302005) to L.G. The funding bodies had no role in the study design, the interpretation of the findings, or the writing of the manuscript.

Conceptualization, Y.L., W.Y., and L.G., Y.X.; methodology, K.H., Y.Z., T.W., L.L., Y.L., Y.W., W.L., L.Z., R.Z., S.Y., H.S., H.D., and L.G.; formal analysis, Y.L., W.Y., and Y.L.; funding acquisition, L.G.; investigation, Y.L., and L.G.; writing: original draft, Y.L., W.Y., and L.G.; and supervision, Q.Y., L.G., and Y.X.

## AUTHOR AFFILIATIONS

[1]Department of Clinical Laboratory, State Key Laboratory of Complex Severe and Rare Diseases, Peking Union Medical College Hospital, Chinese Academy of Medical Sciences and Peking Union Medical College, Beijing, China
[2]Beijing Key Laboratory for Mechanisms Research and Precision Diagnosis of Invasive Fungal Diseases, Beijing, China
[3]Biomedical Engineering Facility of National Infrastructures for Translational Medicine, Institute of Clinical Medicine, Peking Union Medical College Hospital, Chinese Academy of Medical Sciences and Peking Union Medical College, Beijing, China

## AUTHOR ORCIDs

Yingxing Li http://orcid.org/0000-0002-8984-9136
Ying Zhao https://orcid.org/0000-0002-7093-1121
Shuying Yu http://orcid.org/0000-0002-3285-5914
Yingchun Xu http://orcid.org/0000-0002-7126-9459
Lina Guo http://orcid.org/0000-0003-4812-8663

## FUNDING

| Funder | Grant(s) | Author(s) |
| --- | --- | --- |
| MOST \| National Key Research and Development Program of China (NKPs) | 2021YFC2302005 | Lina Guo |

## AUTHOR CONTRIBUTIONS

Yingxing Li, Conceptualization, Data curation, Formal analysis, Investigation, Software, Validation, Visualization, Writing – original draft | Wenhang Yang, Conceptualization, Data curation, Formal analysis, Software, Writing – original draft | Yi Li, Data curation, Formal analysis, Software | Kexin Hua, Methodology | Ying Zhao, Methodology, Resources | Taie Wang, Methodology | Lingli Liu, Methodology | Yali Liu, Methodology, Resources | Yao Wang, Methodology, Resources | Wenjing Liu, Methodology | Li Zhang, Methodology, Resources | Renyuan Zhu, Methodology | Shuying Yu, Methodology | Hongli Sun, Methodology | Hongtao Dou, Methodology | Qiwen Yang, Resources, Supervision | Yingchun Xu, Conceptualization, Resources, Supervision | Lina Guo, Conceptualization, Data curation, Funding acquisition, Investigation, Methodology, Project administration, Resources, Supervision, Writing – original draft

## ETHICS APPROVAL

This study was approved by the Institutional Review Board (IRB) of Peking Union Medical College Hospital, Beijing, China (reference number I-23PJ2169), and informed consent was waived due to the retrospective design of this study.

## ADDITIONAL FILES

The following material is available online.

### Supplemental Material

**Figure S1 (Spectrum02266-24-s0001.docx).** Total number of swabs that tested positive for pathogens and proportion of GBS in vaginal pathogens for each group per year.

### Open Peer Review

**PEER REVIEW HISTORY (review-history.pdf).** An accounting of the reviewer comments and feedback.

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
