## [Reviewer comments · Microbiology Spectrum]

Microbiology Spectrum

The increasing burden of group B *Streptococcus* from 2013 to 2023: a retrospective cohort study in Beijing, China

Yingxing Li, Wenhong Yang, Yi Li, Kexin Hua, Ying Zhao, Taie Wang, Lingli Liu, Ya-li Liu, Yao Wang, Wenjing Liu, Li Zhang, Renyuan Zhu, Shu-Ying Yu, hongli sun, Hongtao Dou, Qiwen Yang, Ying-Chun Xu, and Li-Na Guo

Corresponding Author(s): Yingxing Li, Peking Union Medical College Hospital

Review Timeline:

Submission Date:	September 11, 2024
Editorial Decision:	September 29, 2024
Revision Received:	October 24, 2024
Accepted:	November 8, 2024

Editor: Siu-Kei Chow

Reviewer(s): Disclosure of reviewer identity is with reference to reviewer comments included in decision letter(s). The following individuals involved in review of your submission have agreed to reveal their identity: Sebastian Cifuentes (Reviewer #1)

Transaction Report:

DOI: <https://doi.org/10.1128/spectrum.02266-24>

Re: Spectrum02266-24 (The increasing burden of group B Streptococcus from 2013 to 2023: a retrospective cohort study in Beijing, China)

Dear Dr. Yingxing Li:

Thank you for the privilege of reviewing your work. Below you will find my comments, instructions from the Spectrum editorial office, and the reviewer comments.

Revision Guidelines

Sincerely,
Siu-Kei Chow
Editor
Microbiology Spectrum

Reviewer #1 (Comments for the Author):

Suggestions:

Lines 20-22

Current phrase: GBS is a leading pathogen that can cause fatal infections in newborns due to vertical transmission from the colonized mothers.

Proposed correction: Group B Streptococcus (GBS) is a leading pathogen responsible for fatal infections in newborns, primarily

due to vertical transmission from colonized mothers.

Lines 26-27

Current phrase: Both the GBS detection rate and its proportion among vaginal pathogens indicated a gradual rise in GBS carriage among pregnant and non-pregnant women.

Proposed correction: The detection rate of GBS and its proportion among vaginal pathogens have shown a gradual increase in GBS colonization in both pregnant and non-pregnant women.

Lines 31-32

Current phrase: The number of invasive GBS cases has increased since 2016, particularly among individuals over 40 years old.

Proposed correction: The incidence of invasive GBS cases has risen since 2016, particularly among individuals over the age of 40.

Lines 33-34

Current phrase: The 5,858 GBS isolates we collected exhibited strikingly high resistance rates to erythromycin (71.4%), clindamycin (60%), and levofloxacin (50.1%), with 31.4% being multidrug-resistant.

Proposed correction: The 5,858 GBS isolates exhibited notably high resistance rates to erythromycin (71.4%), clindamycin (60%), and levofloxacin (50.1%), with 31.4% classified as multidrug-resistant.

Lines 35-36

Current phrase: Notably, invasive GBS strains had higher resistance rate to levofloxacin (61.2%) than colonizing strains (49.8%).

Proposed correction: Importantly, invasive GBS strains exhibited a higher resistance rate to levofloxacin (61.2%) compared to colonizing strains (49.8%).

Lines 52-54

Current phrase: Initially, GBS was primarily associated with colonizing the mammary glands of cloven-hoofed animals, causing bovine mastitis and affecting milk yield and quality.

Proposed correction: Initially, GBS was primarily associated with colonization of the mammary glands in cloven-hoofed animals, leading to bovine mastitis and impacting milk yield and quality.

Lines 57-58

Current phrase: GBS is also a major cause of illness and death among infants in both high and low-income countries.

Proposed correction: GBS is a leading cause of morbidity and mortality among infants in both high- and low-income countries.

Lines 63-65

Current phrase: This comprehensive study examined changes in vaginal GBS detection rates among pregnant and non-pregnant women, GBS proportion among vaginal pathogens detected in pregnant and non-pregnant women, clinical characteristics of invasive GBS diseases, and shifts in GBS antibiotic susceptibility profiles from various sample sources.

Proposed correction: This comprehensive study investigated changes in vaginal GBS detection rates among pregnant and non-pregnant women, the proportion of GBS among vaginal pathogens, the clinical characteristics of invasive GBS disease, and shifts in antibiotic susceptibility profiles of GBS isolates from various sample sources.

Lines 77-79

Current phrase: The criteria for defining invasive GBS disease include clinical symptoms of infection associated with GBS isolated from sterile sites such as blood, ascitic fluid, pleural fluid, joint fluid, bronchoalveolar lavage, drainage fluid, etc., as well as skin and soft tissue infections with infection symptoms when GBS is isolated from abscesses, wound secretions, etc.

Proposed correction: The criteria for defining invasive GBS disease include clinical symptoms associated with GBS isolated from sterile sites such as blood, ascitic fluid, pleural fluid, joint fluid, bronchoalveolar lavage, drainage fluid, among others, as well as skin and soft tissue infections where GBS is isolated from abscesses, wound secretions, and other sources.

Lines 333-335

Current phrase: However, we found that its positivity rate was only one-third that of vaginal swabs (data not shown), which aligns with the CDC's recommendation against using PCR as a routine screening method for GBS during delivery.

Proposed correction: "However, we found that its positivity rate was only one-third that of vaginal swabs (data not shown), consistent with the CDC's recommendation against using PCR as a routine screening method for GBS during delivery.

Please ensure that all species names of microorganisms throughout the text are italicized, as per standard scientific conventions. This includes any instances of bacterial, fungal, or other microorganism species mentioned in the manuscript. It would be beneficial to carefully review the document to ensure consistency in formatting for all species names.

Suggestion for Figure 1:

For Figure 1(A), it would be beneficial to include or clarify the sample size (N) associated with each stage of the GBS detection methods. Providing this information would allow readers to assess the robustness of the conclusions drawn about each

method's effectiveness and its evolution over time. Additionally, for Figure 1(B), specifying the sample size for each group (pregnant and non-pregnant women) would strengthen the conclusions about the trends in GBS detection rates and potentially guide future studies on the most effective methods based on statistical significance.

Question to answer:

1. Inconsistencies in N Between Erythromycin and Clindamycin:

In Table 2, we noticed that the number of isolates tested for erythromycin and clindamycin varies significantly across different sample types (e.g., in urine, vaginal, and blood samples). This is unusual if CLSI guidelines were followed, as both erythromycin and clindamycin should have been tested on the same isolates, and the total number of isolates (N) should be consistent for both antibiotics across all sample types. We would like to ask whether all isolates were systematically tested for both antibiotics, and if there was any reason for the differences in N between erythromycin and clindamycin, or if there might have been an error in the data extraction or reporting. Clarifying this point is crucial to ensure consistency and accuracy in the reported results.

2. Definition of Invasive Disease:

We would like to request clarification on how you are defining invasive GBS disease (iGBS) in the study. Are you considering only findings from blood and CSF, or are you including any findings from sterile sites (e.g., pleural, joint fluids, etc.)? It would be beneficial to explicitly define this in the manuscript so that readers clearly understand the scope of the term 'invasive infection.' Additionally, we suggest providing an additional table that describes the antimicrobial resistance profile based on different sites of isolation for invasive infections, along with demographic characteristics (age, comorbidities) of the patients. This would help identify specific patterns in resistance for invasive infections based on age and patient characteristics.

3. Use of Other Antimicrobials:

In the initial text of the manuscript, you mention the use of other antimicrobials for GBS. However, we do not see these antimicrobials included in the results tables, such as Table 2. Could you clarify whether these antimicrobials were evaluated in the study and, if so, provide the resistance results for these drugs? If they were not evaluated, it would be helpful to explicitly state why they were not included in the study, so readers understand whether they were excluded due to a lack of clinical relevance or for other reasons.

4. Clarification on the 'Others' Category in Table 2:

In Table 2, there is a row labeled 'Others' without specifying what sample types it refers to. It would be advisable to clarify exactly what types of samples are included (e.g., rare samples or a grouping of different sample types not categorized in the previous rows). Additionally, it is important to harmonize the data to ensure consistency and avoid confusion. One suggestion would be to break down this 'Others' category into specific sample types, or, if not possible, provide a detailed explanation of what is included in this category. This would improve the clarity of the table and facilitate the interpretation of the results.

5. Penicillin and the 2024 WHO Pathogen Report:

In the recent 2024 WHO Pathogen Report, a growing concern has been identified regarding non-susceptibility of *Streptococcus agalactiae* (GBS) to penicillin. Given that penicillin is a first-line treatment for GBS, we believe it would be highly relevant to include the penicillin susceptibility data in this study. This would allow for an assessment of whether there are circulating GBS strains with resistance or non-susceptibility to penicillin in the studied population. Without this information, it is difficult to fully interpret the antimicrobial resistance landscape, and this could leave a critical gap in the treatment recommendations for invasive infections. We suggest including or clarifying the penicillin susceptibility data, as this is of great importance both clinically and for public health.

6. In the section on antimicrobial susceptibility testing, it would be important to specify the minimum inhibitory concentrations (MICs) for the antibiotics tested and clarify which antibiotics were tested using disk diffusion and which were tested using the broth microdilution method. This distinction is crucial for interpreting the susceptibility results accurately and ensuring alignment with the CLSI guidelines.

Reviewer #2 (Comments for the Author):

Major comments

Drs. Li and Yang et al. performed a retrospective study mainly concerning GBS detection using data from a large tertiary hospital in Beijing from 2013 to 2023. This manuscript contains much information concerning GBS detection. However, because there are many factors which may affect the results, the interpretation of these results is difficult. For example, the duration of this study, from 2013 to 2023, include the duration of COVID-19 pandemic, which strongly affect clinical situations all over the world, maybe including this hospital. Therefore, the interpretation of the results in this manuscript is difficult. Moreover, because this study is based on the data of one institute and this is one of the large limitations of this study, interests of readers in the world may be limited.

Minor comments

Line 52: "*Streptococcus*" and "*Streptococcus agalactiae*" should be typed by the Italic font.

Materials and methods: Because this manuscript contains clinical information, authors should add the Ethical Statement, including the number of the approval of the ethical committee of the institute.

Line 87: Authors should add the name of maker of MALDI-TOF mass spectrometry.

Line 189: *Enterococcus faecalis* is a Gram-positive bacterium.

Line 283: "sensitive" should be "susceptible".

Line 287: Authors should also mention the definition of "multidrug-resistance" at this part.

Figure 1: Authors should show the numbers of isolates of each stage, because stage 3 (2019-2022) is the duration of COVID-19 pandemic.

Figure 3: Authors should show the total numbers of isolates in each year, for the readers to judge the influence of COVID-19 pandemic.

Reviewer #1 (Comments for the Author):

Suggestions:

Response:

We sincerely thank the reviewer for the careful reading of our manuscript and the valuable suggestions for revision. We have revised the following sentences according to the reviewer's recommendations.

Lines 20-22

Current phrase: GBS is a leading pathogen that can cause fatal infections in newborns due to vertical transmission from the colonized mothers.

Proposed correction: Group B Streptococcus (GBS) is a leading pathogen responsible for fatal infections in newborns, primarily due to vertical transmission from colonized mothers.

Lines 26-27

Current phrase: Both the GBS detection rate and its proportion among vaginal pathogens indicated a gradual rise in GBS carriage among pregnant and non-pregnant women.

Proposed correction: The detection rate of GBS and its proportion among vaginal pathogens have shown a gradual increase in GBS colonization in both pregnant and non-pregnant women.

Lines 31-32

Current phrase: The number of invasive GBS cases has increased since 2016, particularly among individuals over 40 years old.

Proposed correction: The incidence of invasive GBS cases has risen since 2016, particularly among individuals over the age of 40.

Lines 33-34

Current phrase: The 5,858 GBS isolates we collected exhibited strikingly high resistance rates to erythromycin (71.4%), clindamycin (60%), and levofloxacin (50.1%), with 31.4% being multidrug-resistant.

Proposed correction: The 5,858 GBS isolates exhibited notably high resistance rates to erythromycin (71.4%), clindamycin (60%), and levofloxacin (50.1%), with 31.4% classified as multidrug-resistant.

Lines 35-36

Current phrase: Notably, invasive GBS strains had higher resistance rate to levofloxacin (61.2%) than colonizing strains (49.8%).

Proposed correction: Importantly, invasive GBS strains exhibited a higher resistance rate to levofloxacin (61.2%) compared to colonizing strains (49.8%).

Lines 52-54

Current phrase: Initially, GBS was primarily associated with colonizing the mammary glands of cloven-hoofed animals, causing bovine mastitis and affecting milk yield and

quality.

Proposed correction: Initially, GBS was primarily associated with colonization of the mammary glands in cloven-hoofed animals, leading to bovine mastitis and impacting milk yield and quality.

Lines 57-58

Current phrase: GBS is also a major cause of illness and death among infants in both high and low-income countries.

Proposed correction: GBS is a leading cause of morbidity and mortality among infants in both high- and low-income countries.

Lines 63-65

Current phrase: This comprehensive study examined changes in vaginal GBS detection rates among pregnant and non-pregnant women, GBS proportion among vaginal pathogens detected in pregnant and non-pregnant women, clinical characteristics of invasive GBS diseases, and shifts in GBS antibiotic susceptibility profiles from various sample sources.

Proposed correction: This comprehensive study investigated changes in vaginal GBS detection rates among pregnant and non-pregnant women, the proportion of GBS among vaginal pathogens, the clinical characteristics of invasive GBS disease, and shifts in antibiotic susceptibility profiles of GBS isolates from various sample sources.

Lines 77-79

Current phrase: The criteria for defining invasive GBS disease include clinical symptoms of infection associated with GBS isolated from sterile sites such as blood, ascitic fluid, pleural fluid, joint fluid, bronchoalveolar lavage, drainage fluid, etc., as well as skin and soft tissue infections with infection symptoms when GBS is isolated from abscesses, wound secretions, etc.

Proposed correction: The criteria for defining invasive GBS disease include clinical symptoms associated with GBS isolated from sterile sites such as blood, ascitic fluid, pleural fluid, joint fluid, bronchoalveolar lavage, drainage fluid, among others, as well as skin and soft tissue infections where GBS is isolated from abscesses, wound secretions, and other sources.

Lines 333-335

Current phrase: However, we found that its positivity rate was only one-third that of vaginal swabs (data not shown), which aligns with the CDC's recommendation against using PCR as a routine screening method for GBS during delivery.

Proposed correction: "However, we found that its positivity rate was only one-third that of vaginal swabs (data not shown), consistent with the CDC's recommendation against using PCR as a routine screening method for GBS during delivery.

Please ensure that all species names of microorganisms throughout the text are italicized, as per standard scientific conventions. This includes any instances of bacterial, fungal, or

other microorganism species mentioned in the manuscript. It would be beneficial to carefully review the document to ensure consistency in formatting for all species names.

Response:

We thank the reviewer for the reminder and apologize for our previous carelessness. We have thoroughly reviewed the entire manuscript to ensure that all species names of microorganisms are correctly written.

Suggestion for Figure 1:

For Figure 1(A), it would be beneficial to include or clarify the sample size (N) associated with each stage of the GBS detection methods. Providing this information would allow readers to assess the robustness of the conclusions drawn about each method's effectiveness and its evolution over time. Additionally, for Figure 1(B), specifying the sample size for each group (pregnant and non-pregnant women) would strengthen the conclusions about the trends in GBS detection rates and potentially guide future studies on the most effective methods based on statistical significance.

Response:

We thank the reviewer for this valuable suggestion. We agree that clarifying the sample size (N) can help readers to assess the robustness of the conclusions. Accordingly, we have updated Figure 1 by adding the total number of swabs submitted for each group per year, allowing readers to easily access the sample sizes across different stages for GBS detection in pregnant and non-pregnant women.

Sample size (N)	Stage 1		Stage 2				Stage 3				Stage 4
	2013	2014	2015	2016	2017	2018	2019	2020	2021	2022	2023
Rectovaginal swab-P			2,090	2,968	2,788	2,912	3,036	2,785	3,102	3,233	3,298
Vaginal swab-P	3,337	3,629	2,739	3,579	3,238	3,392	3,432	3,325	3,467	3,548	3,795
Vaginal swab-NP	3,249	4,253	4,100	4,921	5,184	5,282	4,994	3,625	4,992	5,414	7,551

Figure 1 (A) The evolution of GBS detection method at PUMCH from 2013 to 2023. (B) Upper panel: The trends of vaginal GBS detection rate from pregnant and non-pregnant women in the past decade. Lower panel: The sample size (N) of swabs submitted for each group per year. P indicates pregnant women. NP indicates non-pregnant women.

Question to answer:

1. Inconsistencies in N Between Erythromycin and Clindamycin:

In Table 2, we noticed that the number of isolates tested for erythromycin and clindamycin varies significantly across different sample types (e.g., in urine, vaginal, and blood samples). This is unusual if CLSI guidelines were followed, as both erythromycin and clindamycin should have been tested on the same isolates, and the total number of isolates (N) should be consistent for both antibiotics across all sample types. We would like to ask whether all isolates were systematically tested for both antibiotics, and if there was any reason for the differences in N between erythromycin and clindamycin, or if there might have been an error in the data extraction or reporting. Clarifying this point is crucial to ensure consistency and accuracy in the reported results.

Response:

We thank the reviewer for raising this concern. Our clinical antimicrobial susceptibility testing has consistently followed the CLSI guidelines. However, our study is retrospective, with data sourced from routine clinical antimicrobial testing, including some changes in testing methods between 2013 and 2023, as detailed in Table 1. Specifically, from 2013 to 2017, we used the disk diffusion method for all antibiotics, and the number of GBS isolates tested for erythromycin and clindamycin susceptibility was roughly the same. Between 2018 and 2020, our hospital introduced the VITEK 2 AST-P639 card (bioMérieux, Marcy l'Etoile, France), based on broth microdilution, for routine susceptibility testing of Gram-positive bacteria. During this period, most GBS isolates were tested using this system. However, the VITEK 2 AST-P639 card has certain limitations, as it does not provide results for chloramphenicol and clindamycin susceptibility in GBS. The difference in the number of isolates tested for erythromycin and clindamycin mainly arises from the use of this testing card during these three years. From 2021 to 2023, our hospital resumed using the disk diffusion method for most GBS isolates, resulting in similar numbers of isolates tested for erythromycin and clindamycin susceptibility during this period. The performance of the VITEK 2 AST-P639 card in susceptibility testing has been clinically evaluated and shown to have accuracy comparable to the disk diffusion method, making the results from both methods equally reliable.

Table R1 The sample size for the specific antibiotic susceptibility testing methods used each year for clinically isolated GBS strains during 2013-2023.

Sample size	Total	Erythromycin		Chloramphenicol		Clindamycin		Levofloxacin	
		ND*	NM**	ND	NM	ND	NM	ND	NM
Y-2013	216	213	0	213	0	211	0	214	0
Y-2014	310	302	0	304	0	305	0	308	0
Y-2015	388	341	0	337	0	338	0	385	0
Y-2016	501	419	0	425	0	427	0	497	0
Y-2017	517	421	0	422	0	430	0	489	0
Y-2018	495	157	251	161	0	169	0	199	296
Y-2019	608	20	515	19	0	21	0	20	587
Y-2020	496	161	290	134	0	167	0	172	322

Y-2021	732	614	39	618	0	627	0	640	93
Y-2022	708	601	28	605	0	613	0	628	80
Y-2023	887	731	22	744	0	740	0	767	120

ND*: Disk diffusion method.

NM**: Broth microdilution method using VITEK 2 AST-P639 card.

2. Definition of Invasive Disease:

We would like to request clarification on how you are defining invasive GBS disease (iGBS) in the study. Are you considering only findings from blood and CSF, or are you including any findings from sterile sites (e.g., pleural, joint fluids, etc.)? It would be beneficial to explicitly define this in the manuscript so that readers clearly understand the scope of the term 'invasive infection.' Additionally, we suggest providing an additional table that describes the antimicrobial resistance profile based on different sites of isolation for invasive infections, along with demographic characteristics (age, comorbidities) of the patients. This would help identify specific patterns in resistance for invasive infections based on age and patient characteristics.

Response:

We thank the reviewer for this suggestion. In the Materials and Methods section, we have described the criteria for defining invasive GBS disease as follows. “The criteria for defining invasive GBS disease include the isolation of GBS from normally sterile sites such as blood, abdominal fluid, pleural fluid, joint fluid, bronchoalveolar lavage, drainage fluid or cerebrospinal fluid. It also involves skin and soft tissue infections, where GBS is detected in abscesses, pus and wound secretions, accompanied by local or systemic signs and symptoms of inflammation.”

Following the reviewer’s suggestion, we added a table to represent the antibiotic resistance profile of GBS isolates from patients with invasive infections, categorized by different age groups, underlying conditions, and clinical manifestations (Table 3). From Table 3, we can observe some interesting patterns. For example, the resistance rate of GBS to levofloxacin increases with the age of patients with invasive GBS infections. Additionally, the lowest levofloxacin resistance rate is observed in skin or soft-tissue infection, while the highest resistance rate occurs in bacteremia without focus. Considering that resistance rates can fluctuate greatly and lack statistical significance when the sample size is too small (fewer than 10 cases), we only selected representative underlying conditions and clinical manifestations with more than 10 GBS isolates for analysis. We have added Table 3 and related content in the Results section of the revised manuscript (line 327-333).

Table 3 The antibiotic resistance rates of invasive GBS isolates categorized by representative clinical characteristics of patients

Resistant rate	Erythromycin	Chloramphenicol	Clindamycin	Levofloxacin
Total	75.2% (121/161)	8.9% (7/79)	69.3% (61/88)	61.3% (100/163)
Age group				
Neonates	100% (3/3)	0% (0/2)	66.7% (2/3)	66.7% (2/3)
16-39 years	67.5% (27/40)	15.8% (3/19)	71.4% (15/21)	55% (22/40)

40-64 years	80% (56/70)	8.8% (3/34)	73.7% (28/38)	59.2% (42/71)
≥ 65 years	72.9% (35/48)	4.2% (1/24)	61.5% (16/26)	69.4% (34/49)
Underlying condition				
≥1 condition	74.2% (92/124)	6.7% (4/60)	71.2% (47/66)	65.1% (82/126)
Cancer	77.8% (49/63)	6.7% (2/30)	77.4% (24/31)	62.5% (40/64)
Cardiovascular disease	71.4% (40/56)	4.3% (1/23)	65.4% (17/26)	66.1% (37/56)
Diabetes mellitus	65.1% (28/43)	5.3% (1/19)	59.1% (13/22)	65.9% (29/44)
Autoimmune disease	87.0% (20/23)	8.3% (1/12)	100% (13/13)	60.9% (14/23)
Liver disease	77.8% (14/18)	20.0% (1/5)	83.3% (5/6)	66.7% (12/18)
Peripheral vascular disease	71.4% (10/14)	16.7% (1/6)	100% (6/6)	64.3% (9/14)
Clinical manifestations				
Skin or soft-tissue infection	75.4% (49/65)	14.3% (5/35)	73.7% (28/38)	55.4% (36/65)
Bacteremia without focus	73.7% (28/38)	13.3% (2/15)	73.7% (14/19)	69.2% (27/39)
Intra-abdominal infection	73.7% (14/19)	0% (0/9)	60% (6/10)	63.2% (12/19)
Pneumonia	72.7% (8/11)	0% (0/8)	62.5% (5/8)	66.7% (8/12)

3. Use of Other Antimicrobials:

In the initial text of the manuscript, you mention the use of other antimicrobials for GBS. However, we do not see these antimicrobials included in the results tables, such as Table 2. Could you clarify whether these antimicrobials were evaluated in the study and, if so, provide the resistance results for these drugs? If they were not evaluated, it would be helpful to explicitly state why they were not included in the study, so readers understand whether they were excluded due to a lack of clinical relevance or for other reasons.

Response :

We thank the reviewer for pointing out this problem. In addition to erythromycin, clindamycin, levofloxacin, and chloramphenicol, we also tested most clinical GBS isolates for resistance to penicillin G (N=5,729), ceftriaxone (N=4,290), linezolid (N=5,831), and vancomycin (N=5,769) in our study. However, we did not find any isolates non-susceptible to any of these four antibiotics. This content has been described in line 296-298 of the revised manuscript.

4. Clarification on the 'Others' Category in Table 2:

In Table 2, there is a row labeled 'Others' without specifying what sample types it refers to. It would be advisable to clarify exactly what types of samples are included (e.g., rare samples or a grouping of different sample types not categorized in the previous rows). Additionally, it is important to harmonize the data to ensure consistency and avoid confusion. One suggestion would be to break down this 'Others' category into specific sample types, or, if not possible, provide a detailed explanation of what is included in this category. This would improve the clarity of the table and facilitate the interpretation of the results.

Response :

We thank the reviewer for this valuable suggestion. Previously, we categorized rare

samples with fewer than 20 cases and uncertain samples from the medical records system under the "Other" category. Following the reviewer's suggestion, we now list all sample types, including rare ones, and group only uncertain samples into the "Not Determined" category. The updated Table 2 and related analysis are presented below. "An analysis of sample types with more than 10 cases reveals the following antibiotic resistance patterns: erythromycin resistance is higher in tissue (91.7%), throat (83.3%), and secretion (81.5%) samples; chloramphenicol resistance is elevated in throat (18.2%), pus (15.4%), and wound (13.6%) samples; clindamycin resistance is higher in throat (90.9%), wound (77.3%), and secretion (75%) samples; and levofloxacin resistance is more prevalent in blood (66.7%), throat (66.7%), and drainage (65.7%) samples. Notably, GBS isolates from throat samples exhibit high resistance levels across all four antibiotics, whereas those from rectovaginal swabs show resistance rates lower than the overall average. As previously mentioned, GBS strains responsible for invasive infections display significantly higher levofloxacin resistance than colonizing strains. Further examination of levofloxacin resistance across sample types reveals that isolates from sterile sites—such as blood, drainage, pleural fluid, abdominal fluid, and joint fluid—exhibit resistance rates exceeding 60%, markedly higher than the overall average resistance rate (50.1%)." The new Table 2 and the corresponding analysis have been added to the revised manuscript in the results section at line 315-326.

Table 2 The antibiotic resistance rates of GBS isolates categorized by specimen types

Resistance rate	Erythromycin	Chloramphenicol	Clindamycin	Levofloxacin
Total	72.2% (3698/5121)	9.5% (379/3982)	60.0% (2431/4050)	50.1% (2910/5810)
Vagina	73.7% (2245/3046)	10.6% (262/2471)	60.7% (1517/2501)	50.4% (1543/3064)
Rectovaginal swab	70.0% (1142/1637)	7.7% (96/1246)	57.5% (725/1261)	43.6% (718/1648)
Urine	64.2% (52/81)	5.9% (4/68)	54.9% (39/71)	60.5% (446/737)
Sputum	69.4% (34/49)	14.7% (5/34)	68.4% (26/38)	54.9% (28/51)
Blood	70.2% (33/47)	11.1% (2/18)	58.3% (14/24)	66.7% (32/48)
Drainage	65.7% (23/35)	0% (0/19)	63.2% (12/19)	65.7% (23/35)
Wound	74.3% (26/35)	13.6% (3/22)	77.3% (17/22)	57.1% (20/35)
Pus	79.3% (23/29)	15.4% (2/13)	73.3% (11/15)	48.3% (14/29)
Secretion	81.5% (22/27)	5.6% (1/18)	75% (15/20)	64.3% (18/28)
Skin	66.7% (14/21)	0% (0/10)	63.6% (7/11)	47.6% (10/21)
Semen	73.3% (11/15)	0% (0/9)	33.3% (3/9)	26.7% (4/15)
Rectal (neonates)	66.7% (12/18)	0% (0/9)	66.7% (6/9)	43.8% (7/16)
Throat	83.3% (10/12)	18.2% (2/11)	90.9% (10/11)	66.7% (8/12)
Tissue	91.7% (11/12)	0% (0/4)	50% (2/4)	41.7% (5/12)
Mouth (neonates)	54.5% (6/11)	0% (0/2)	66.7% (2/3)	58.3% (7/12)
Tracheobronchial aspirate	100% (8/8)	0% (0/3)	100% (4/4)	50% (4/8)
Pleural fluid	60% (3/5)	0% (0/4)	50% (2/4)	66.7% (4/6)
Abdominal fluid	60.0% (3/5)	0% (0/2)	100.0% (3/3)	60.0% (3/5)
Broncho-alveolar	100% (4/4)	0% (0/2)	100% (2/2)	100% (4/4)

lavage				
Catheter	100% (2/2)	0% (0/1)	100% (1/1)	100% (2/2)
Joint fluid	100% (2/2)	0% (0/1)	100% (1/1)	100% (2/2)
Breast milk	50% (1/2)	0% (0/2)	100% (2/2)	50% (1/2)
Ear	50% (1/2)	0% (0/1)	100% (1/1)	0% (0/2)
Umbilical cord (neonates)	50% (1/2)	0% (0/2)	50% (1/2)	50% (1/2)
Not determined	58.3% (7/12)	25% (2/8)	70% (7/10)	41.7% (5/12)

5. Penicillin and the 2024 WHO Pathogen Report:

*In the recent 2024 WHO Pathogen Report, a growing concern has been identified regarding non-susceptibility of *Streptococcus agalactiae* (GBS) to penicillin. Given that penicillin is a first-line treatment for GBS, we believe it would be highly relevant to include the penicillin susceptibility data in this study. This would allow for an assessment of whether there are circulating GBS strains with resistance or non-susceptibility to penicillin in the studied population. Without this information, it is difficult to fully interpret the antimicrobial resistance landscape, and this could leave a critical gap in the treatment recommendations for invasive infections. We suggest including or clarifying the penicillin susceptibility data, as this is of great importance both clinically and for public health.*

Response:

We are grateful to the reviewer for raising this important question. Since penicillin is a first-line treatment for GBS, routine susceptibility testing for penicillin is performed on all clinical GBS isolates. In this study, we assessed the penicillin susceptibility of 5,729 GBS isolates, with 1,488 tested using VITEK 2 AST-P639 card (broth microdilution method) and 4,251 using the disk diffusion method (10 isolates were tested with both methods). Given the importance of monitoring penicillin resistance in GBS isolates, any non-susceptible strains detected during clinical routine testing undergo a comprehensive re-evaluation. This process consists of three steps: (1) purifying the isolate, (2) confirming species identification, and (3) re-testing antibiotic susceptibility. After re-evaluation, we found no penicillin-resistant strains among the 5,729 clinical GBS isolates. However, as most of our routine testing relies on the disk diffusion method, it is difficult to monitor trends in the MIC values for penicillin. In future studies, we will continue to focus on penicillin resistance in GBS isolates and plan to increase the use of the broth microdilution method to better track changes in MIC values over time.

6. In the section on antimicrobial susceptibility testing, it would be important to specify the minimum inhibitory concentrations (MICs) for the antibiotics tested and clarify which antibiotics were tested using disk diffusion and which were tested using the broth microdilution method. This distinction is crucial for interpreting the susceptibility results accurately and ensuring alignment with the CLSI guidelines.

Response:

We thank the reviewer for pointing out this problem. We have provided all relevant details and added them to the Materials and Methods section of the revised manuscript. "GBS isolates were further characterized to determine the antimicrobial susceptibility profile. The

antibiotic susceptibility of GBS to penicillin G, ceftriaxone, vancomycin, linezolid, erythromycin, clindamycin, chloramphenicol, and levofloxacin was assessed using either the disk diffusion method or the broth microdilution method, following the CLSI M100 ED34 guidelines. The broth microdilution method was performed using the VITEK 2 AST-P639 card (bioMérieux, Marcy l'Etoile, France), an antibiotic susceptibility testing system specifically designed for Gram-positive bacteria. Both the disk diffusion and broth microdilution methods were employed to evaluate the susceptibility of isolates to penicillin (susceptible: zone diameter ≥ 24 mm or MIC ≤ 0.12 $\mu\text{g/ml}$), vancomycin (susceptible: zone diameter ≥ 17 mm or MIC ≤ 1 $\mu\text{g/ml}$), linezolid (susceptible: zone diameter ≥ 21 mm or MIC ≤ 2 $\mu\text{g/ml}$), erythromycin (resistant: zone diameter ≤ 15 mm or MIC ≥ 1 $\mu\text{g/ml}$), and levofloxacin (resistant: zone diameter ≤ 13 mm or MIC ≥ 8 $\mu\text{g/ml}$). For ceftriaxone, clindamycin, and chloramphenicol, only the disk diffusion method was used, with susceptibility or resistance thresholds defined as follows: ceftriaxone (susceptible: zone diameter ≥ 24 mm), clindamycin (resistant: zone diameter ≤ 15 mm), and chloramphenicol (resistant: zone diameter ≤ 17 mm)."

Reviewer #2 (Comments for the Author):

Major comments

Drs. Li and Yang et al. performed a retrospective study mainly concerning GBS detection using data from a large tertiary hospital in Beijing from 2013 to 2023. This manuscript contains much information concerning GBS detection. However, because there are many factors which may affect the results, the interpretation of these results is difficult. For example, the duration of this study, from 2013 to 2023, include the duration of COVID-19 pandemic, which strongly affect clinical situations all over the world, maybe including this hospital. Therefore, the interpretation of the results in this manuscript is difficult. Moreover, because this study is based on the data of one institute and this is one of the large limitations of this study, interests of readers in the world may be limited.

Response :

We thank the reviewer for bringing this insightful concern to our attention. The potential relationship between changes in GBS detection rates and the COVID-19 pandemic is indeed a fascinating issue that we had previously overlooked. In response, we have now incorporated a relevant analysis into the Discussion section of the revised manuscript (line 394-407). Based on two pieces of evidence—GBS detection rates and the proportion of GBS among vaginal pathogens—we inferred that the vaginal colonization rates of GBS in both pregnant and non-

pregnant women have been increasing from 2013 to 2023, with a more significant rise in pregnant women. Our analysis suggests that the impact of the COVID-19 pandemic on GBS carriage rates may differ between these two groups.

For pregnant women, the GBS carriage rate had already significantly increased before the onset of the pandemic in 2019, and this upward trend continued through 2023, after the pandemic ended. In fact, the GBS carriage rate in 2023 exceeded that of the pandemic years (2020-2022). Therefore, we speculate that the increase in GBS carriage rates in pregnant women is unlikely to be directly caused by the COVID-19 pandemic. However, in the non-pregnant group, there was no significant rise in GBS carriage rates before the pandemic in 2020. During the three pandemic years (2020-2022), the GBS carriage rate in non-pregnant women significantly increased, but after the pandemic ended in 2023, it slightly declined, though it remained higher than pre-pandemic levels. Thus, we hypothesize that the COVID-19 pandemic may have contributed to the increase in GBS carriage in non-pregnant women. Nevertheless, more data in the coming years will be needed to infer the precise impact of the pandemic on GBS carriage rates, and we will continue to monitor changes in GBS detection rates in the future.

Minor comments

Line 52: "Streptococcus" and "Streptococcus agalactiae" should be typed by the Italic font.

Response :

We are grateful to the reviewer for pointing this out and sincerely apologize for our carelessness. We have corrected this error and thoroughly reviewed all species names of microorganisms throughout the manuscript to ensure they are correctly italicized.

Materials and methods: Because this manuscript contains clinical information, authors should add the Ethical Statement, including the number of the approval of the ethical committee of the institute.

Response :

We thank the reviewer for pointing out this problem. The Ethical Statement has been included in the Materials and methods of the revised manuscript as follows. "This study was proved by the Institutional Review Board (IRB) of Peking Union Medical College Hospital, Beijing, China (reference number I-23PJ2169), and the informed consent was waived due to the retrospective design of this study."

Line 87: Authors should add the name of maker of MALDI-TOF mass spectrometry.

Response :

We thank the reviewer for this question. The name of maker of MALDI-TOF mass spectrometry (bioMérieux, Marcy l'Etoile, France) has been included in the Materials and Methods section of the revised manuscript.

Line 189: Enterococcus faecalis is a Gram-positive bacterium.

Response:

We thank the reviewer for the careful reading of our manuscript. We apologize for this error and have made the correction.

Line 283: "sensitive" should be "susceptible".

Response:

The word "sensitive" has been replaced by "susceptible" in the revised manuscript (line 297).

Line 287: Authors should also mention the definition of "multidrug-resistance" at this part.

Response:

We thank the reviewer for raising this question. The "multidrug-resistance" at this part is defined as GBS being resistant to erythromycin, clindamycin, and levofloxacin simultaneously (line 300-302).

Figure 1: Authors should show the numbers of isolates of each stage, because stage 3 (2019-2022) is the duration of COVID-19 pandemic.

Response:

We thank the reviewer for this suggestion. In response, we have revised Figure 1 by adding the total number of swabs (sample size) submitted for each group per year. From these data (Figure R1A), we observe that the number of vaginal or rectovaginal swabs submitted for pregnant women between 2020 and 2022 remained largely unaffected by the COVID-19 pandemic. This suggests that the increase in GBS detection rates during this period was not due to changes in sample volume. To further clarify our findings, we have also included the annual number of GBS-positive cases (Figure R1B) and the yearly GBS detection rates (Figure R1C). Notably, GBS detection rates from both vaginal and rectovaginal swabs in pregnant women increased significantly in 2019, well before the onset of the COVID-19 pandemic. This indicates that the observed rise in GBS detection rates was unrelated to the pandemic. Moreover, this upward trend persisted through 2023, even after the pandemic had ended, reinforcing the conclusion that the increase in vaginal GBS colonization among pregnant women is likely independent of COVID-19-related disruptions.

In contrast, the number of swabs submitted for non-pregnant women was slightly impacted by the pandemic, with a decline in 2020 due to China's strict isolation and control measures, followed by an increase in 2023 when no such measures were in place. Interestingly, the number of vaginal swabs from non-pregnant women in 2021 and 2022 remained comparable to pre-pandemic levels. Excluding the data from the pandemic period (2020-2022), we find that the GBS detection rates in 2019 and 2023 are still higher than the average levels from 2013 to 2018. Based on these findings, we conclude that the GBS colonization rate among non-pregnant women is also gradually increasing, although the rate of increase is smaller than that observed among pregnant women.

A

Sample size (N)	Stage 1		Stage 2				Stage 3				Stage 4
	2013	2014	2015	2016	2017	2018	2019	2020	2021	2022	2023
Rectovaginal swab-P			2,090	2,968	2,788	2,912	3,036	2,785	3,102	3,233	3,298
Vaginal swab-P	3,337	3,629	2,739	3,579	3,238	3,392	3,432	3,325	3,467	3,548	3,795
Vaginal swab-NP	3,249	4,253	4,100	4,921	5,184	5,282	4,994	3,625	4,992	5,414	7,551

B

Number of GBS isolates	Stage 1		Stage 2				Stage 3				Stage 4
	2013	2014	2015	2016	2017	2018	2019	2020	2021	2022	2023
Rectovaginal swab-P			126	176	149	145	241	236	322	323	350
Vaginal swab-P	64	71	81	71	68	66	114	104	141	147	178
Vaginal swab-NP	103	171	174	218	211	207	216	165	272	286	373

C

GBS detection rate (%)	Stage 1		Stage 2				Stage 3				Stage 4
	2013	2014	2015	2016	2017	2018	2019	2020	2021	2022	2023
Rectovaginal swab-P			6.03	5.93	5.34	4.98	7.94	8.47	10.38	9.99	10.61
Vaginal swab-P	1.92	1.96	2.96	1.98	2.10	1.95	3.32	3.13	4.07	4.14	4.69
Vaginal swab-NP	3.17	4.02	4.24	4.43	4.07	3.92	4.33	4.55	5.45	5.28	4.94

Figure R1 The total number of swabs collected (A), the number of GBS positive cases (B) and GBS detection rate (%) (C) for each group per year.

Figure 3: Authors should show the total numbers of isolates in each year, for the readers to judge the influence of COVID-19 pandemic.

Response:

We thank the reviewer for this suggestion. Figure R2 shows the total number of vaginal swabs tested positive for pathogens (Figure R2A) and proportion of GBS in vaginal pathogens (%) (Figure R2B) per year. We have included Figure R2 in the supplemental materials for the readers to judge the influence of COVID-19 pandemic.

A

Total number of swabs tested positive for pathogens	2013	2014	2015	2016	2017	2018	2019	2020	2021	2022	2023
Vaginal swab-P	632	733	533	571	459	553	624	506	574	540	693
Vaginal swab-NP	806	1167	1071	1289	1290	1276	1317	885	1235	1279	2057

B

Proportion of GBS in vaginal pathogens (%)	2013	2014	2015	2016	2017	2018	2019	2020	2021	2022	2023
Vaginal swab-P	10.127	9.6862	15.197	12.434	14.815	11.935	18.269	20.553	24.564	27.222	25.685
Vaginal swab-NP	12.779	14.653	16.247	16.912	16.357	16.223	16.401	18.644	22.024	22.361	18.133

Figure R2 The total number of swabs tested positive for pathogens (A) and proportion of GBS in vaginal pathogens (%) (B) for each group per year.

Re: Spectrum02266-24R1 (The increasing burden of group B *Streptococcus* from 2013 to 2023: a retrospective cohort study in Beijing, China)

Dear Dr. Yingxing Li:

Your manuscript has been accepted, and I am forwarding it to the ASM production staff for publication. Your paper will first be checked to make sure all elements meet the technical requirements. ASM staff will contact you if anything needs to be revised before copyediting and production can begin. Otherwise, you will be notified when your proofs are ready to be viewed.

Sincerely,
Siu-Kei Chow
Editor
Microbiology Spectrum